# GRA12 is a common virulence factor across *Toxoplasma gondii* strains and mouse subspecies

Francesca Torelli[1,2], Simon Butterworth [1], Eloise Lockyer[1], Ana N. Matias [2], Franziska Hildebrandt[2], Ok-Ryul Song [3], Jennifer Pearson-Farr[4] & Moritz Treeck [1,2] ✉

*Toxoplasma gondii* parasites exhibit extraordinary host promiscuity owing to over 250 putative secreted proteins that disrupt host cell functions, enabling parasite persistence. However, most of the known effector proteins are specific to *Toxoplasma* genotypes or hosts. To identify virulence factors that function across different parasite isolates and mouse strains that differ in susceptibility to infection, we performed systematic pooled in vivo CRISPR-Cas9 screens targeting the *Toxoplasma* secretome. We identified several proteins required for infection across parasite strains and mouse species, of which the dense granule protein 12 (GRA12) emerged as the most important effector protein during acute infection. GRA12 deletion in IFNγ-activated macrophages results in collapsed parasitophorous vacuoles and increased host cell necrosis, which is partially rescued by inhibiting early parasite egress. GRA12 orthologues from related coccidian parasites, including *Neospora caninum* and *Hammondia hammondi*, complement TgΔGRA12 in vitro, suggesting a common mechanism of protection from immune clearance by their hosts.

*T oxoplasma gondii* is one of the most prevalent parasites worldwide and infects any nucleated cell of warm-blooded animals, including humans[1]. Most isolates can be grouped into three lineages – type I, II, and III – with progressively decreasing virulence in mice[2]. Less virulent type II and III strains dominate in Europe and North America[3], and infection is generally asymptomatic in humans. Less common but more virulent type I strains have been associated with clinical complications in humans, such as visual impairment and foetal malformations[4,5]. In South America, particularly in Brazil, a multitude of genetically diverse isolates have been identified over the last decades[6]. These so-called "atypical" isolates[7] caused several lethal outbreaks affecting immunocompetent individuals, high miscarriage rates and lack of cross-protective immunity[8,9]. While work of the last decades largely focussed on identifying virulence factors that make one strain more virulent than another, the factors conserved across strains and conferring *Toxoplasma* success in a wide host range remain unknown to date.

*Toxoplasma*'s capacity to escape immune clearance is largely conferred by a pool of proteins secreted from two secretory organelles during and after invasion: rhoptries (ROPs) and dense granules (GRAs). Secreted proteins generate a supporting environment for *Toxoplasma* growth by rewiring host cell transcription and interfering with the host cell immune machinery[10]. For example, the dense granule inhibitor of STAT1 transcriptional activity (IST) blocks the upregulation of interferon-stimulated genes in naïve cells and subsequent parasite clearance[11–13], while the rhoptry kinase ROP16 drives macrophage polarisation to support parasite growth[14,15]. Secreted proteins either reside within the non-fusogenic parasitophorous vacuole (PV) created

[1]Signalling in Apicomplexan Parasites Laboratory, The Francis Crick Institute, London, UK. [2]The Cell Biology of Host-Pathogen Interactions Lab, Gulbenkian Institute for Molecular Medicine, Lisbon, Portugal. [3]High-Throughput Screening Technology Platform, The Francis Crick Institute, London, UK. [4]Electron Microscopy Science Technology Platform, The Francis Crick Institute, London, UK. ✉e-mail: moritz.treeck@gimm.pt

by the actively growing intracellular tachyzoites, span the PV membrane (PVM) or cross the PVM into the host cytoplasm. Recently, the number of secreted proteins has been estimated to total around 250, the vast majority without functional annotations[16].

To date, the most extensively characterised virulence factor is the serine-threonine kinase ROP18, which is directly associated with parasite virulence[17,18]. ROP18, together with the pseudokinase ROP5, inactivates the main murine defence mechanism against intracellular pathogens: the Immunity-Related GTPases (IRGs)[17,19]. The IRG family has specifically expanded in the murine genome[20], where it includes 23 members, and has been associated with host resistance to intracellular pathogens of parasitic, bacterial and fungal origin[21–23]. Specifically, IRGa6, IRGb6, IRGd and IRGb10, are pivotal for host survival to *Toxoplasma* infection via coordinated loading on the PVM[24], which depends on the regulatory IRGs, namely IRGm1 and IRGm3[25,26]. Irgm1/m3 are also paramount for the Ubiquitin-mediated degradation of the *Toxoplasma* vacuole and for general cellular homeostasis[27,28]. In human cells, IRGs are almost entirely missing. Another family of GTPases, the guanylate binding proteins (GBPs), is involved in *Toxoplasma* restriction in both human and murine hosts. mGBP1 and mGBP2 control *Toxoplasma* infection in murine cells[29,30], and hGBP1, hGBP2 and hGBP5 in human macrophages[31]. Recent work showed that murine resistance to *Toxoplasma* relies on the PV nitrosylation and collapse of GBP2-recruited vacuoles[32]. IRGs and GBPs loading on the PVM results in membrane ruffling, vacuolar rounding and breakage eventually resulting in parasite elimination via an unknown mechanism[24,29,30]. Parasite clearance leads to host cell death, which is considered a hallmark of host resistance to infection[33,34]. The activation of specific programmed host cell death pathways, like apoptosis and pyroptosis[35,36], were observed following loading of IRGs and GBPs.

All main mouse subspecies – i.e. *M. m. domesticus*, *M. m. musculus* and *M. m. castaneus* – can control infection with the less virulent type II and III Toxoplasma strains, however the outcomes after infection with the highly virulent type I parasites are different. For example, infection with type I strains is lethal in the most commonly used *M. m. domesticus* mouse strains, e.g. C57BL/6J, where ROP18 inhibits the vacuolar accumulation of host IRGa6[19]. Type II and III parasites express less virulent isoforms or undetectable levels of ROP18 respectively[37]. Due to polymorphisms in the IRG loci, *M. m. castaneus* and *M. m. musculus* strains are resistant to ROP18 function and survive infection[33,38,39]. Therefore, ROP18 is a virulence factor, but only in some murine subspecies and parasite strains. Most South American *Toxoplasma* strains are lethal in all three subspecies[38,39], suggesting the presence of unknown parasite virulence factors in this *Toxoplasma*-host combination.

Most virulence factors characterised to date are specific to certain clonal strains, host species or cell types, or a combination of them[40]. This is likely biased by the initial studies to identify strain-specific virulence factors based on *Toxoplasma* genetic crosses, and by the predominant use of *M. m. domesticus* laboratory mouse strains as model organisms[41]. This is of particular importance when considering the key role of rodents in *Toxoplasma* transmission, as they are major prey of felines, the definitive host of the parasite. Thus, whilst strain- and species-specific virulence factors have been identified, we do not know which virulence factors are important in all parasite strains to colonise different hosts. If they exist, these conserved effectors significantly contribute to *Toxoplasma* promiscuity and abundance in nature.

To identify common virulence factors acting across mouse strains of different genetic backgrounds, we performed targeted CRISPR-Cas9 screens in three *Toxoplasma* strains belonging to the common lineages and the atypical VAND strain, in different murine subspecies. The dense granule protein 12 (GRA12) emerged as the most important secreted protein for survival in the mouse peritoneum during acute infection and in IFNγ-activated macrophages, irrespective of parasite

or mouse genetic backgrounds. Deletion of GRA12 led to increased regulated necrotic host cell death, partially caused by parasite early egress. A collapsed vacuolar space indicated a function of GRA12 in maintaining this replicative niche during the IFNγ-mediated host cell defence. Complementation of GRA12 deletion with GRA12 orthologues from two closely related parasite species suggested that the protective mechanism is conserved and important beyond *Toxoplasma* infections.

## Results

### Targeted in vivo CRISPR screens identify GRA12 as a *Toxoplasma* strain-transcendent secreted virulence factor in the mouse peritoneum

To identify virulence factors that function across parasite strains and mouse species, we generated pooled CRISPR mutant libraries as previously described[42–44] in RH ΔHXGPRT and PRU ΔHXGPRT (hereafter named RH and PRU), VEG and VAND parasites to encompass type I, type II, type III and atypical strains, respectively. Parasite knock-out (KO) pools for 253 predicted rhoptry and dense granule proteins were injected intraperitoneally (i.p.) into five mice per parasite strain (Fig. 1a). To avoid skewing the results towards the known strain-specific virulence factor ROP18 in susceptible *M. m. domesticus* mice, we used selected mouse subspecies for the CRISPR screens: for the more virulent VAND and RH strains we used the *M. m. castaneus* CAST/EiJ and *M. m. musculus* PWD/PhJ, respectively (called "IRG resistant mice"). The different subspecies were used in these two screens due to loss of mouse colonies during the COVID-19 pandemic. The PRU and VEG screens were performed in *M. m. domesticus* C57BL/6J (called "IRG susceptible mice"). Five days post infection, parasites were recovered from the peritoneal exudates, their sgRNAs amplified by PCR and sequenced to determine their relative abundance before and after in vivo selection (Fig. 1a). Genes that contribute to survival in vivo will display a negative log$_2$ fold change (L2FC), because parasites lacking those genes will be cleared during the infection.

We observed comparable variability, in terms of median absolute deviation (MAD) of the L2FC and sgRNA loss, in vitro and in vivo in three of the four screens. The VEG screen displayed a higher variability (Supplementary Fig. 1a–d), which is probably promoted by the strong tendency of VEG to encyst[45]. Encystation leads to growth arrest, which likely increases the dropout rate of mutants from the pool and reduction of valid data points. Regardless, in vitro and in vivo L2FC scores correlate well with previous screens performed in our laboratory in the PRU strain (Supplementary Fig. 1e, f). Data points with a more pronounced difference between the median L2FC in vivo and in vitro display a higher discordance/concordance (DISCO) score which is adjusted for the *p*-value of each parameter (Fig. 1b). All raw and normalised read counts, L2FC and DISCO scores for each screen are reported in Supplementary Data 3–6. To identify virulence factors common between strains, we ranked the candidates based on the difference between L2FC in vivo and in vitro (L2FC DIFF, Supplementary Data 7) using the cumulative score for each candidate across all screens (Table 1). The relatively high loss of mutants in the VEG screen led to a loss of data points for most genes identified as important in the other *Toxoplasma* strains. However, relaxing the inclusion criteria from three guides per gene to one guide per gene for the VEG screen resulted in data points for all ranked virulence factors (Supplementary Fig. 1g and Table 1). While the values from the VEG screen should be taken with some caution, they indicate that several proteins play important roles across strains.

The dense granule protein GRA12 is most strongly correlated with reduced fitness upon deletion across all screens, regardless of the mouse or parasite backgrounds (Fig. 1b and Table 1). In contrast, and as expected, correlation between ROP18 deletion and parasite fitness depends on parasite and mouse genotypes. Another important protein identified across the different screens is GRA45, which is

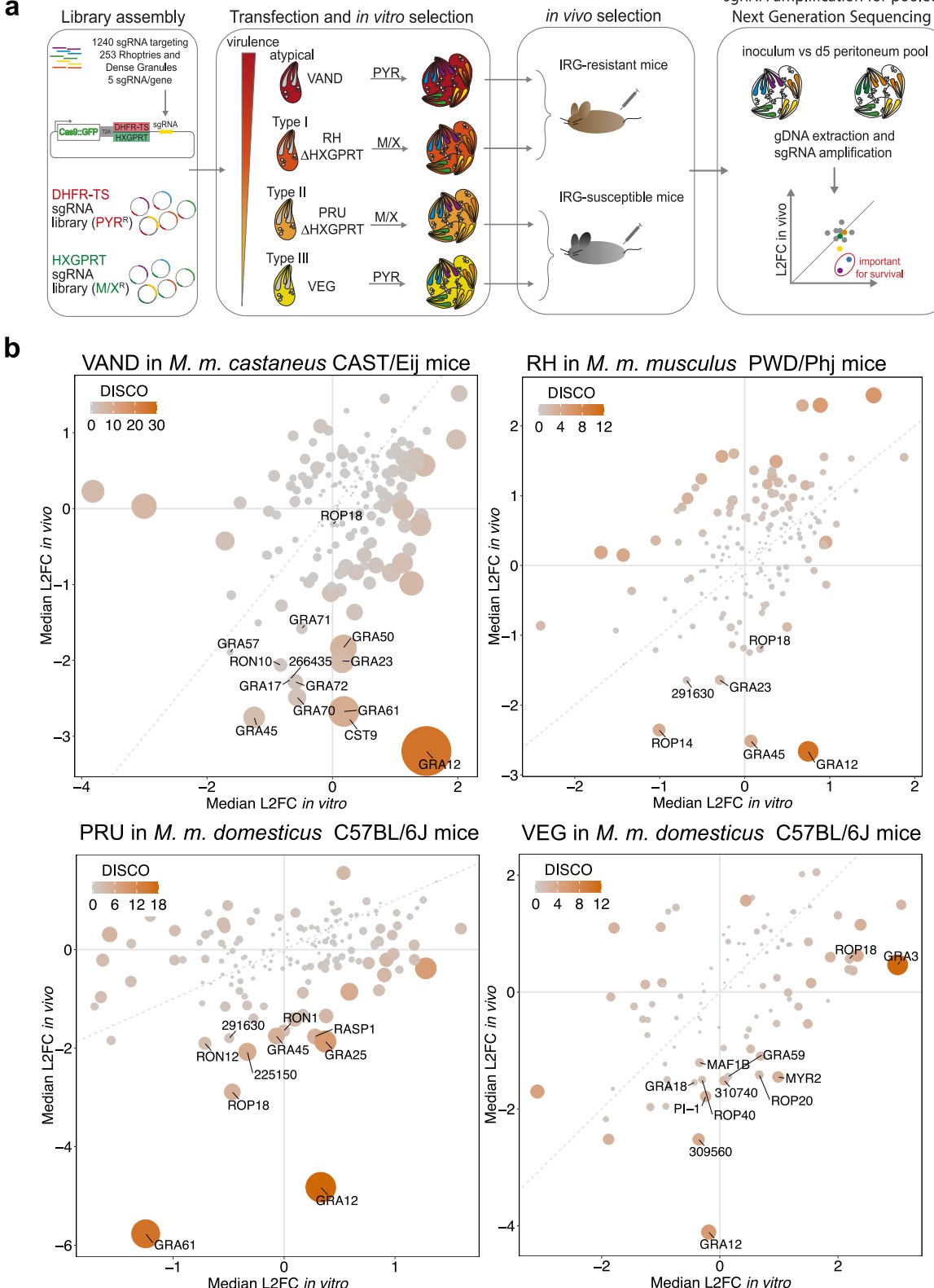

involved in the correct localisation of proteins in the PVM[46], one of which, GRA23, was also identified in our screens[47]. Furthermore, GRA61 and GRA50 have been identified to be fitness conferring in vivo in all performed CRISPR screens to date[42,48–50], but their functions remain unknown.

The screens also suggest that novel strain-specific virulence factors exist, such as the predicted dense granule protein TGGT1_291630

orthologs in PRU and RH, or orthologs of the rhoptry proteins TGGT1_309560 and TGGT1_266435 in VEG and VAND respectively. Interestingly, three proteins (GRA57, GRA70 and GRA71) previously identified as important factors for *Toxoplasma* survival in human cells[44,51], are uniquely identified in the VAND screen, suggesting a strain-specific role of this complex in this parasite and mouse combination that deserves further investigation.

**Fig. 1 | CRISPR screens in vivo identify GRA12 as the most important common secreted virulence factor for survival across clonal and atypical *Toxoplasma* strains. a** Scheme of the CRISPR screen pipeline. Two protospacers libraries targeting the putative *Toxoplasma* secretome were assembled and included either the DHFR-TS or the HXGPRT selection cassette to create pyrimethamine-resistant (PYR[R]) or mycophenolic acid/xanthine-resistant (M/X[R]) parasite KO pools. The wild-type strains VAND and VEG were transfected with the DHFR-TS library, while the ΔHXGPRT strains RH and PRU were transfected with the HXGPRT library. The resulting KO parasite pools were injected in the peritoneum of 5 mice/parasite strain, retrieved after 5 days and expanded for one lytic cycle in vitro before gDNA

extraction and sgRNA amplification for sequencing. The relative abundance of each guide (L2FC) at day 5 versus inoculum indicates the importance for in vivo survival of the relative *Toxoplasma* gene. **b** Scatter plots of the median L2FC for each gene in vitro and in vivo in CRISPR screens of, in order from the top left: VAND in CAST/EiJ mice, RH ΔHXGPRT in PWD/PhJ mice, PRU ΔHXGPRT and VEG in C57BL/6J mice. The colour and size of each point reflects the Discordance/Concordance (DISCO) score, and the dashed grey line indicates equal L2FC. Displayed data are thresholded for L2FC in vitro > −1 according to published in vitro whole genome screens[53]. Data points with a L2FC in vivo < −1.5 are labelled in addition to ROP18 as control.

**Table 1 | The five top common virulence factors across clonal and atypical *Toxoplasma* strains**

| Rank | Gene | Code | VAND | | RH | | PRU | | VEG | |
|---|---|---|---|---|---|---|---|---|---|---|
| | | | L2FC in vivo | DISCO | L2FC in vivo | DISCO | L2FC in vivo | DISCO | L2FC in vivo | DISCO |
| 1 | GRA12 | _288650 | −3.20 | 26.04 | −2.66 | 10.73 | −4.82 | 17.71 | −4.11 | 5.87 |
| 2 | GRA45 | _316250 | −2.75 | 4.55 | −2.52 | 4.01 | −1.75 | 4.89 | −5.25 (1)* | 5.25* |
| 3 | GRA61 | _269950 | −2.67 | 9.52 | −0.41 | 0.36 | −5.77 | 15.77 | −3.70 (2)* | 4.11* |
| 4 | GRA50 | _203600 | −1.84 | 7.08 | 0.32 | 1.72 | −0.21 | 3.53 | −2.47 (2)* | 4.31* |
| 5 | GRA23 | _297880 | −2.01 | 5.66 | −1.64 | 2.10 | −0.16 | 0.42 | −3.93 (1)* | 8.50* |

Genes were ranked based on the difference between the median L2FC in vivo and in vitro across all strains. The median L2FC in vivo and the DISCO score are reported for the five top hits. For the VEG screen numbers are reported also from the quantification with relaxed selection (>=1 guide/ gene). Values obtained only with relaxed selection are indicated with an asterisk, with the number of guides used for quantification in brackets.

While we cannot exclude that we missed some virulence factors because of the variability of the VEG screen, GRA12 emerges as the most critical virulence factor between all tested parasite strains in all mouse genetic backgrounds. GRA12 has been previously identified in in vivo CRISPR screens performed in type I and II strains by our research group and others[42,48–50] and is important for the survival of IFNγ-mediated restriction in vitro[52] and in vivo[49]. However, the importance of GRA12 in other strains, such as type III and atypical, and how it mediates *Toxoplasma* survival had not been previously characterised.

### GRA12 is a virulence factor in the South American VAND strain in vivo

To validate GRA12 as a virulence factor in IRG resistant mice, we generated a ΔGRA12 and a GRA12-complemented strain in the atypical South American VAND parasite strain. The endogenous *Gra12* locus was replaced with a mCherry expression cassette via CRISPR-Cas9 and loss of the GRA12 expression was validated by immunofluorescence detection with an anti-GRA12 antibody (Supplementary Fig. 2a, b). An HA-tagged complemented strain (GRA12::HA) was established via homologous recombination in the *Uprt* locus (Supplementary Fig. 2c). Western blot analysis confirmed a GRA12::HA band at the expected size of 48 kDa, and GRA12::HA was localised to the intravacuolar space by immunofluorescence (Fig. 2a, b). In line with an in vitro fitness score of +1.75[53], lack of GRA12 confers a modest growth advantage to *Toxoplasma* in human foreskin fibroblasts (HFFs) and mouse embryonic fibroblasts (MEFs) in plaque assays, which is lost in the complemented strain (Supplementary Fig. 2d). Interestingly, plaque size was generally smaller in MEFs compared to HFFs. Intraperitoneal infection of IRG resistant PWD/PhJ mice with a dose of 50 to 500 parental and GRA12-complemented parasites was lethal, while mice infected with up to hundred times higher doses of ΔGRA12 parasites survived (Fig. 2c). The surviving mice presented with VAND ΔGRA12 cysts in the brain, showing that GRA12 is not essential for latent stage formation in vivo (Fig. 2d). However, as wildtype (WT)-infected mice succumb to infection before cyst formation, we cannot draw conclusions about the conversion rate. In conclusion, we identified GRA12 as a common virulence factor and validated its importance in clonal and atypical strains.

### GRA12 is essential to survive the IFNγ-mediated clearance in murine macrophages of different genetic backgrounds

Macrophages are the major cell type infected in the peritoneum during the acute phase[15]. To identify genes specifically important for *Toxoplasma* survival in macrophages, we performed a CRISPR screen using the type I RH KO pool in PWD/PhJ mice bone marrow-derived macrophages (BMDMs) with or without IFNγ pretreatment (Fig. 3a). We used the RH pool to prevent the spontaneous encystation commonly seen for VAND, and PWD/PhJ mice to validate parasite factors required in IRG resistant mice. All raw and normalised read counts, L2FC and DISCO scores are reported in Supplementary Data 8 and the sgRNA dropout rate at each passage in Supplementary Fig. 3a. In line with the in vivo screens, GRA12 deletion resulted in the highest fitness defect (Fig. 3b). While the strong phenotype observed for GRA12 deletion may have skewed the other effectors phenotypes towards a more neutral score, GRA61, TGGT1_309560, TGGT1_243690, ROP18 and GRA45 also appear to play a role in macrophage survival, but to a lesser extent.

We next validated the results in macrophage restriction assays using newly generated knockout strains of UPRT (which mimic WT conditions), ROP18 and GRA12, together with the GRA12::HA complemented parasite line. All strains were validated by PCR and IFA or western blot (Supplementary Fig. 3b–h). Plaque assays showed the previously observed increased growth for parasites lacking GRA12, and plaques were smaller in MEFs compared to HFFs (Supplementary Fig. 3d). The complemented GRA12 parasite line did not show the expected rescue of the plaque size. However, since GRA12 function is fully restored in these parasites in IFNγ restriction assays, the lack of rescue is likely caused by a secondary mutation in the parasites and was not further pursued here. Restriction assays were performed using high-content imaging of BMDMs from *M. m. domesticus* C57BL/6J mice, *M. m. musculus* PWD/PhJ mice and *M. m. castaneus* CAST/EiJ mice, and parasite burden was quantified on the cumulative mCherry signal of mutant parasites following previously established protocols[43,44]. At 24 h post infection (hpi), ΔGRA12 parasites were consistently cleared more than parental strains in IFNγ-primed conditions, which was rescued in the GRA12-complemented line (Fig. 3c and Supplementary Fig. 4a). As expected, ROP18 plays no role in conferring protection against the IFNγ response in CAST/EiJ BMDMs (Fig. 3c and

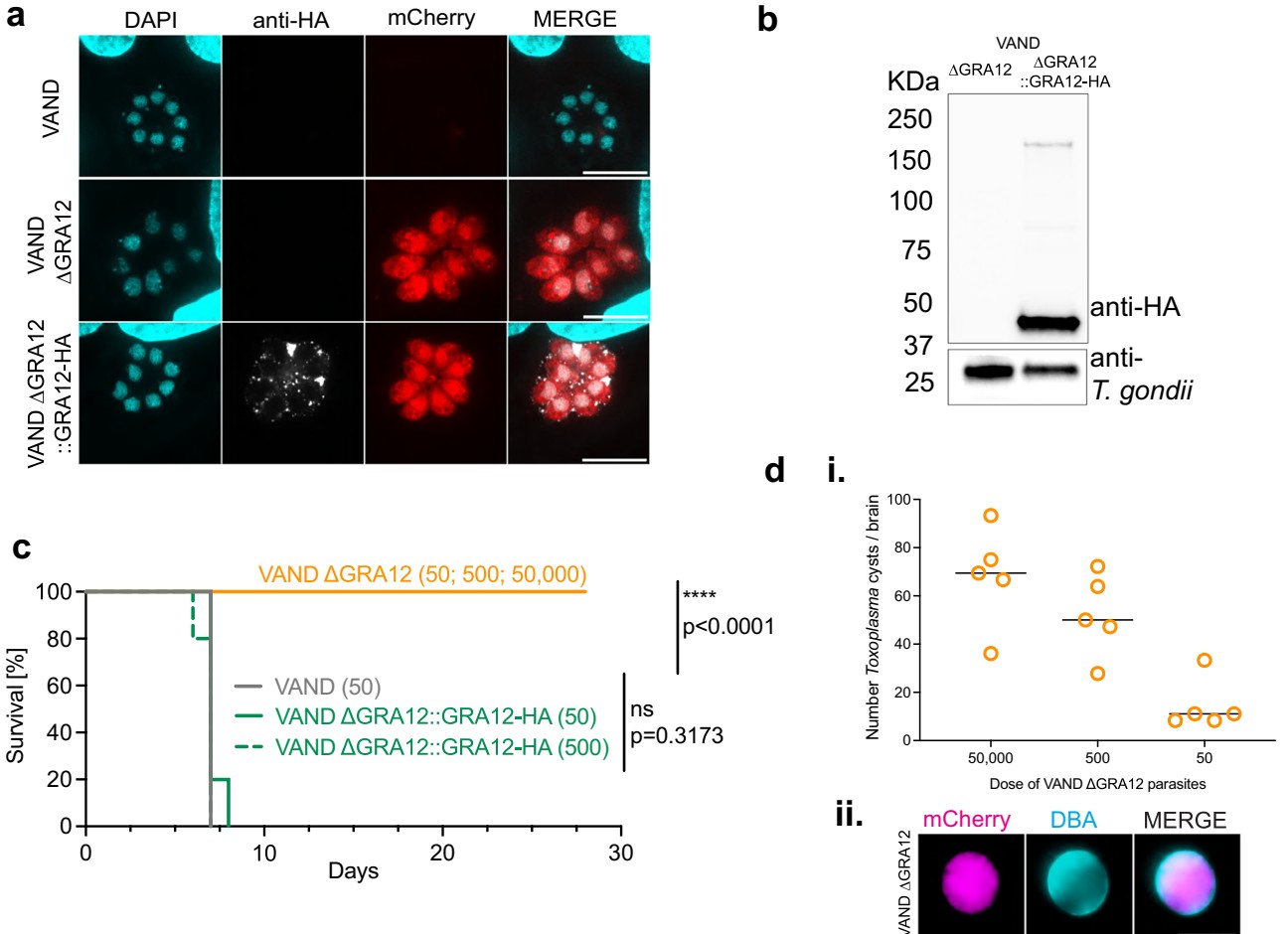

Fig. 2 | GRA12 is a virulence factor in the highly virulent South American VAND strain in vivo. a Immunofluorescence localisation of the C-terminal HA-tagged GRA12 in the VAND ΔGRA12::GRA12-HA strain. Scale bar represents 10 μm. b Verification of GRA12 expression by anti-HA detection in the VAND ΔGRA12::-GRA12-HA strain via western blot. c Survival curve of PWD/PhJ mice infected with either VAND, ΔGRA12 or ΔGRA12::GRA12-HA strains, dose in parenthesis.

Significance was tested using a two-sided Mantel-Cox test, $N = 5$ mice per group. d Number of brain cysts recovered from PWD/PhJ mice infected with VAND ΔGRA12 parasites, $N = 5$. (i) and immunofluorescence detection of a representative brain cyst stained with DBA. Scale bar represents 50 μm (ii). Source data are provided as a Source Data file.

Supplementary Fig. 4a). ROP18 contributed to parasite survival in PWD/PhJ BMDMs, albeit to a lesser extent than GRA12. This result matches a ROP18 intermediate phenotype observed in the CRISPR screen in PWD/PhJ in vivo, but is in contrast with previous work[39]. Contrary to previously published data, this mouse strain was also susceptible to type I GT1 strain infection (Supplementary Fig. 4b). Sanger sequencing of the PWD/Phj mice confirmed that the IRG genes known to be important for parasite restriction (*Irgb6*, *Irgm1*, *Irgb2-b1*) carry the resistant alleles, indicating that an as yet unidentified factor in these mice confers susceptibility to ROP18 expressing parasites. Given that the phenotypic behaviour diverges from published data, results in the PWD/PhJ mouse strain warrant further investigation, but is beyond the scope of this study.

To explore GRA12 function across different host species, we performed restriction assays in Wistar rat BMDMs and human foreskin fibroblasts, but no significant change was observed (Supplementary Figs. 4c, d and[44,51]). As previously reported, GRA12 contributes to IFNγ-mediated parasite survival in murine fibroblasts, although to a lesser extent than in BMDMs (Supplementary Fig. 4d and 52). Overall, our data indicate that GRA12 is the first virulence factor validated in different mouse subspecies and across parasite strains, but has probably no function under these conditions in rats or humans.

## Lack of GRA12 results in host cell death by regulated necrosis following *Toxoplasma* infection

To assess if deletion of GRA12 leads to an increase in host cell death upon IFNγ treatment, we measured the uptake of cell impermeable propidium iodide into the host cell over time. Macrophage cell death was increased in cells infected with ΔGRA12 parasites of either VAND, RH and PRU strains at 9 hpi when compared to controls (15–34%) and rescued by the parental and GRA12-complemented strains (Fig. 4a, b and Supplementary Fig. 5a, b).

Based on the increased cell death in GRA12 deletion mutants, we investigated which cell death pathway is activated upon ΔGRA12 infection. A ~4-fold increase of the necrosis-associated high mobility group B1 protein marker (HMGB1) was found in the supernatant of ΔGRA12 infected macrophages compared to controls (Fig. 4c, quantification in Supplementary Fig. 5c). Cells did not display features of apoptosis or necroptosis, as indicated by lack of cleaved caspase 8 and phosphorylation of MLKL respectively (Fig. 4c). Live time-lapse microscopy of IFNγ-treated BMDMs infected with parasites showed two main cell death phenotypes: a sudden shrinkage followed by a "burst", or blebbing with nuclear condensation, with parasites sometimes egressing from dying cells. Both phenotypes appeared regardless of the strain used (Fig. 4d, Supplementary Movie 1). Numerous

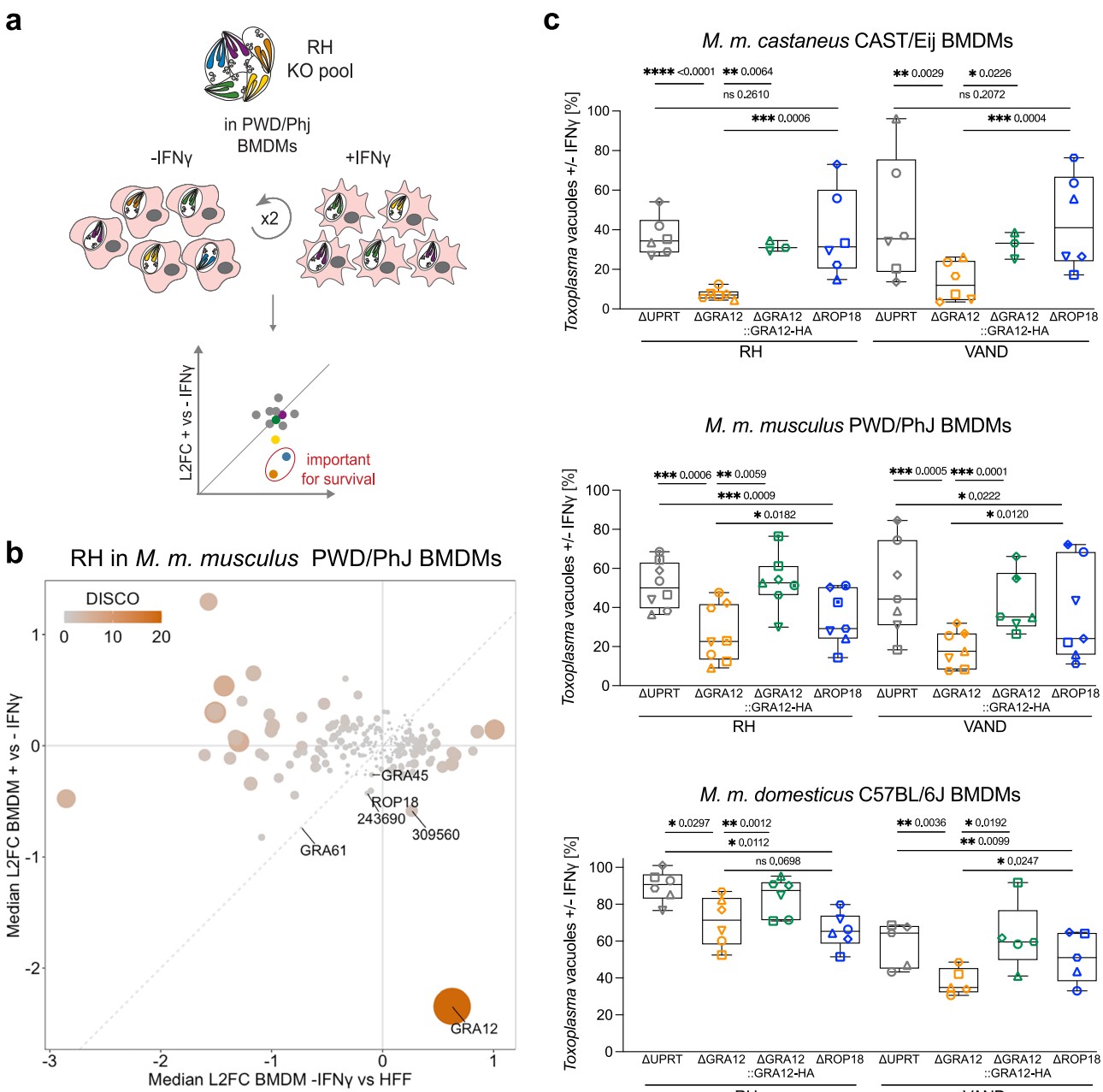

**Fig. 3 | GRA12 is the first identified virulence factor to survive the IFNγ-mediate clearance in different murine subspecies. a** Scheme of the CRISPR screen of type I RH secretome in PWD/PhJ BMDMs to identify *Toxoplasma* factors important to survive IFNγ-mediated restriction. The KO pool was used to infect BMDMs either untreated or pretreated with IFNγ for two consecutive lytic cycles. Surviving parasites were expanded in fibroblasts prior to gDNA extraction and sgRNA amplification for sequencing. The relative abundance of each guide (L2FC) in IFNγ-restricted versus untreated parasites indicates the importance for in vitro survival of the relative *Toxoplasma* gene. **b** Scatter plots of the median L2FCs for each gene in vitro in CRISPR screens of the RH KO pool in PWD/PhJ BMDMs, treated or not

with IFNγ. The grey line indicates equal L2FCs. **c** Quantification of high content-automated imaging of parasite vacuoles in IFNγ-treated BMDMs relative to untreated controls. BMDMs of different mouse subspecies were infected with RH or VAND ΔUPRT, ΔGRA12, ΔGRA12::GRA12-HA and ΔROP18 and vacuoles were quantified at 24 h after infection. The box-plot shows the median value ± SD and the whiskers show minimum and maximum values. Symbol shapes indicate biological repeats, *N* > = 3. Significance was tested with the One-way Anova test with the Benjamini, Krieger and Yekutieli FDR correction. Source data are provided as a Source Data file.

extracellular parasites were observed specifically in IFNγ-treated macrophages infected with ΔGRA12 parasites (Supplementary Fig. 5d). Therefore, we hypothesised that the lack of GRA12 might promote host cell necrosis caused by early egress of the parasite. Consequently, we treated parasites with a cGMP-dependent protein kinase (PKG) inhibitor developed for *Plasmodium falciparum* parasites, ML10[54]. ML10 treatment blocked parasite egress (Fig. 4e), indicating

that the compound also functions in *Toxoplasma*. ML10 treatment reduced parasite numbers, but did not interfere with IFNγ-mediated restriction (Supplementary Fig. 5e). ML10 treatment partially reduced the propidium iodide uptake in ΔGRA12 infected macrophages in a IFNγ-dependent manner (Fig. 4f, g and Supplementary Fig. 5f), suggesting that early egress of ΔGRA12 parasites contributes to increased necrotic cell death, but is not the sole driver.

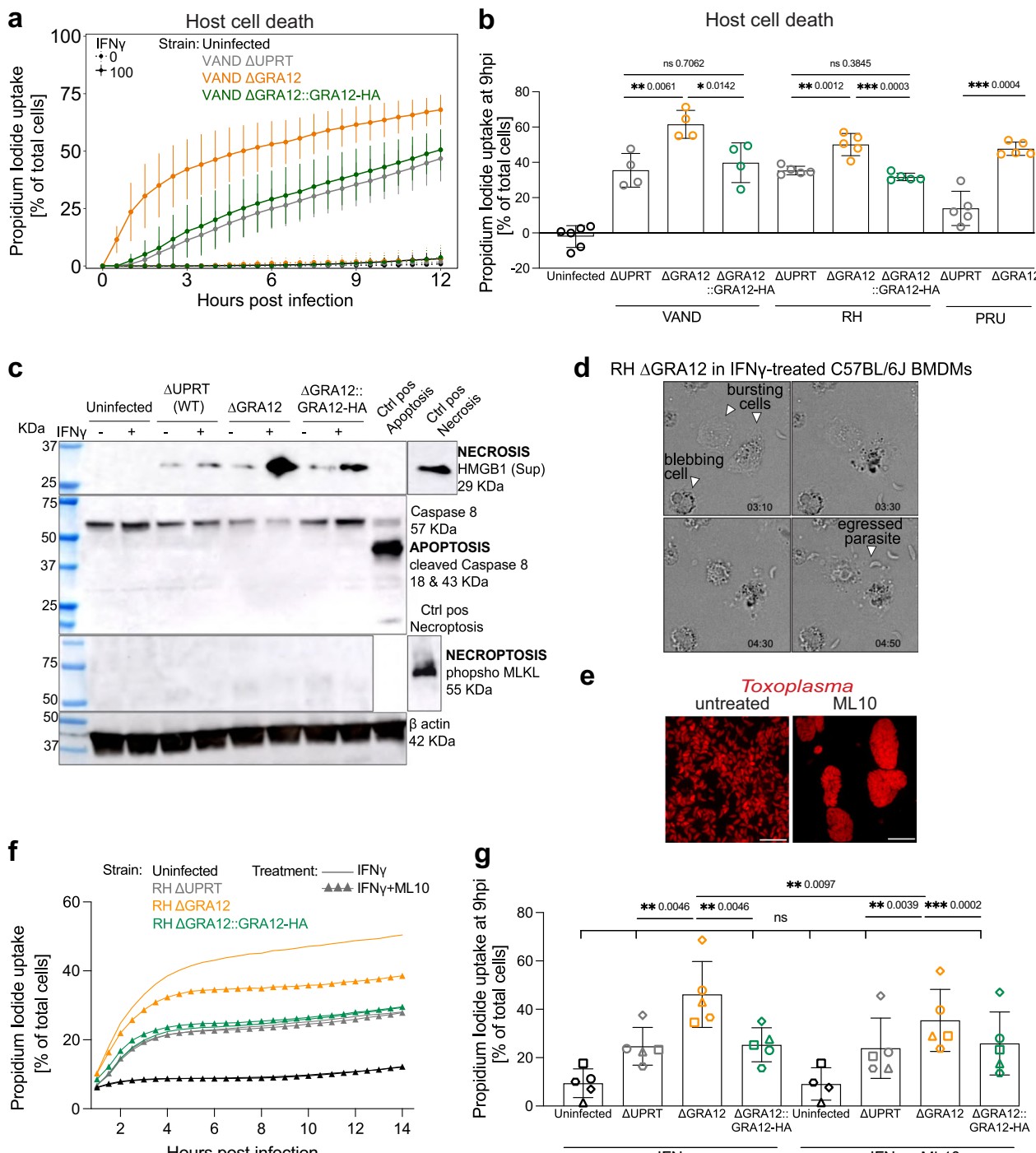

**Fig. 4 | Lack of GRA12 results in IFNγ-mediated host cell death by necrosis.**
**a** Time course of PWD/PhJ BMDMs cell death quantified via Propidium Iodide uptake. The graph shows the median value ± SEM, $N = 4$. Cells were pre-stimulated for 24 h with IFNγ (continuous line) or left untreated as control (dashed line) and then infected with VAND ΔUPRT, ΔGRA12, ΔGRA12::GRA12-HA or left uninfected. The number of dead cells is expressed as a percentage of the total at each time point. **b** Cell death quantification at 9 hpi of IFNγ-treated PWD/PhJ BMDMs infected with ΔUPRT, ΔGRA12, ΔGRA12::GRA12-HA in the PRU, RH and VAND strains, or left uninfected as control. The bar graph shows the median value ± SD, $N > = 4$. Significance was tested using the Tukey's multiple comparisons test for the RH and VAND strains, and with a paired t test for the PRU strains. **c** Western blot analysis of the supernatant and cell lysate of C57BL/6J BMDMs pre-treated with IFNγ and infected for 8 h with RH ΔUPRT, ΔGRA12, ΔGRA12::GRA12-HA, or left untreated as control. **d** Live cell imaging of IFNγ-treated C57BL/6J BMDMs infected with RH

ΔGRA12 parasites. Time post infection reported as hh:mm in the bottom right corner. Filled arrowheads indicate dying cells and an egressed parasite. Complete video is in Supplementary Movie 1. **e** Fluorescence microscopy detection of RH ΔUPRT parasites after 48 h growth in HFFs treated or not with 1 μM ML10. Scale bar represents 20 μm. **f** Time course of C57BL/6J BMDMs cell death quantified via Propidium Iodide uptake. Cells were pre-stimulated for 24 h with IFNγ, and infected with RH ΔUPRT, ΔGRA12, ΔGRA12::GRA12-HA before treatment with 1 μM ML10 (triangle symbol), or left untreated as control (continuous line). The number of dead cells is expressed as a percentage of the total at each time point. **g** Cell death quantification at 9 hpi of the ML10 treatment experiment. The bar graph shows the median value ± SD, $N > = 4$. Symbol shapes indicate biological repeats. Significance was tested using the One-way Anova test with the Benjamini, Krieger and Yekutieli FDR correction. Source data are provided as a Source Data file.

**Parasites lacking GRA12 have a collapsed intravacuolar space and are less targeted by host immune proteins**

GRA12 is predicted to contain a signal peptide, a short N-terminus, a putative transmembrane domain, and a larger 36 KDa C-terminal domain[55]. As previously observed for human cells, GRA12 localises within the vacuole in murine cells (Supplementary Fig. 6a), where it predominantly colocalises with the IVN marker GRA2, and only partially with the PVM marker GRA3 (Supplementary Fig. 6b)[52,55]. Interestingly, its localisation appeared more clustered in fibroblasts than murine macrophages. No differential localisation of GRA12 was observed upon IFNγ treatment (Supplementary Fig. 6a) and the absence of GRA12 did not affect GRA2 and GRA3 localisation in murine macrophages treated or not with IFNγ (Supplementary Fig. 6c). To test if the N- or C-terminus may face the host cell cytoplasm where they could directly interact and interfere with host cell proteins, we assessed the topology of GRA12. We generated an endogenously C-terminally HA-tagged line (RH GRA12-HA) and a complemented N-terminally HA-tagged line (RH ΔGRA12::HA-GRA12, Supplementary Fig. 6d–f). Both strains complement the growth increase observed for GRA12 deletion strains in plaque assays and the IFNγ phenotype in restriction assays (Supplementary Fig. 6g, h). We used low concentrations of the mild detergent saponin to permeabilise only a fraction of PVMs, and localise the GRA12 termini in either the cytoplasmic or vacuolar compartments via immunofluorescence detection with an anti-HA antibody. HA signal was only detected in vacuoles that were also anti-*Toxoplasma* antibody positive, indicating that both ends of GRA12 face the luminal side of the PV and therefore do not face the host cell cytosol (Fig. 5a).

To identify putative GRA12 interaction partners, we performed protein co-immunoprecipitation (co-IP) in IFNγ-treated murine macrophages infected with GRA12-HA or wildtype parasite as control and analysed the samples by mass spectrometry. As expected, GRA12 showed the highest enrichment (Supplementary Fig. 7a, b and Supplementary Data 9), but only a few putative host interactors were identified: the murine ATPase plasma membrane Ca2+ transporting 1 (ATP2B1) and the mitochondrial protein Prohibitin 2 (PHB2). However, control IPs and immunofluorescence assays failed to confirm their specific enrichment at the vacuole or reciprocal co-immunoprecipitation with GRA12. Previously identified interactors of GRA12, such as GRA2, ROP5 and ROP18[55,56] were not identified in this experiment. These may have been missed here because of technical differences between pull down protocols. But given that GRA12 is important even in strains where ROP5 and ROP18 have no protective role, these interactions are probably not important for GRA12 function.

To assess whether lack of GRA12 induces changes in the vacuolar area not visible by immunofluorescence, we performed Transmission Electron Microscopy (TEM) of infected BMDMs. Parasites lacking GRA12 have significantly reduced vacuolar space in IFNγ-treated macrophages compared to the untreated condition and in over 30% of cases the PVM is not clearly separated from the parasite plasma membrane (Fig. 5b–d and Supplementary Fig. 7c). Vacuole collapse has been associated with nitric oxide (NO) production and GBP2 recruitment[32]. However, infection with ΔGRA12 parasites neither increases NO levels (Supplementary Fig. 7d), nor is restriction rescued by the iNOS inhibitor 1400W (Supplementary Figs. 7e, f). We also observed a 2–6 fold decrease in IRGd, IRGb10 and GBP2 recruitment to the PV in PRU ΔGRA12 infected cells compared to parental (Fig. 5e, f and Supplementary Fig. 8a), and a similar, but not significant trend for Ubiquitin recruitment (Supplementary Fig. 8b). Recruitment in the type I RH strain was overall low as expected[52,57], but a significant decrease was observed for IRGd and IRGb10 (Supplementary Fig. 8c–f). In summary, these data suggest that GRA12 is an entirely intravacuolar protein and its deletion leads to collapsed vacuoles and reduced presence of host immune factors on the vacuole.

**GRA12 homologues from closely related Coccidian parasites rescue *Toxoplasma* ΔGRA12 restriction**

GRA12 has four paralogues (GRA12A-D) which share an overall pairwise amino acid sequence identity below 30% and that are, except for GRA12C, highly expressed in both tachyzoite and bradyzoite stages[58]. All Coccidia parasites have 1 to 4 *Gra12*-like orthologues (Fig. 6a). We and others have shown that GRA12 has a key role for *Toxoplasma* survival in vivo[48–50], while the paralogues GRA12A, GRA12B and GRA12D contribute to virulence only during mild infections and are linked to cyst formation[58].

The *Hammondia hammondi* and *Neospora caninum* orthologues closest to *Toxoplasma* GRA12 share 85% and 53% of the amino acid sequence respectively. The GRA12 protein structure prediction by Alphafold suggests a conserved fold, even with the more diverse *Neospora* homologue (Fig. 6b). This suggests a similar function of GRA12 in these related species. To test this hypothesis, we complemented RH ΔGRA12 with HA-tagged orthologues of *Hammondia hammondi* (HHA_288650) or *Neospora caninum* (Ncaninum_LIV_000214200, Supplementary Fig. 9a, b). Both proteins localised within the vacuole, similar to the *Toxoplasma* GRA12 (TgGRA12, Supplementary Fig. 9c) and rescued IFNγ-mediated restriction of RH ΔGRA12 parasites in restriction assays (Fig. 6c). This suggests that GRA12 also functions to prevent the IFNγ-mediated clearance in closely related Coccidian parasites.

## Discussion

*Toxoplasma* can infect and persist in a wide range of hosts. While past research has focused on secreted virulence factors that render some strains more pathogenic than others, no protein that allows the parasite to persist in all hosts has been identified yet. Here, using systematic in vivo CRISPR-Cas9 screens, we identified secreted virulence factors that function across *Toxoplasma* strains and mouse genotypes. Among these, deletion of GRA12 consistently had the most pronounced effect on parasite survival in vivo.

We show that the absence of GRA12 leads to a rapid death of the infected cell. Early egress partially explains the increase in host cell death, and time-lapse imaging suggests that it is probably not preceding it. Parasite egress is more likely a response to the rapid induction of the host immune response, causing the activation of regulated cell death pathways, cellular shrinkage and changes in intracellular ion concentrations[59], triggering egress. The observed cell death phenotypes do not resemble cell death mechanisms previously associated to *Toxoplasma* infection, such as pyroptosis and necroptosis, which are characterised by an initial swelling phase[60]. This indicates that probably a different cell death pathway is induced. While cell death phenotypes have been observed in both wild type and knock out parasites, an increased frequency of events was registered in ΔGRA12 infected cells both by propidium iodide uptake and live cell imaging. We did not observe or test the ability of egressed parasites to successfully infect neighbouring cells. However, the strong reduction of the ΔGRA12 parasites both in vivo and in vitro suggests that the combined effect of host niche destruction and enhanced parasite egress is sufficient to halt the infection. Since GRA12 does not seem to span the membrane to interact with host proteins, and we did not identify convincing protein interaction partners, we hypothesise that GRA12 might have a role in stabilising the PVM. As a result, the deletion of GRA12 might establish hypersensitivity to clearance mechanisms, such as vacuolar breakage, increasing egress and triggering of host-cell death pathways. The collapsed vacuolar space of ΔGRA12 parasites associated with a decreased recognition from host immune proteins would support this hypothesis, similarly to what was previously observed for deletion strains of GRA57, GRA70 and GRA71 when infecting human cells[44,51].

Strikingly, GRA12 orthologues from closely related parasites rescued GRA12 function, indicating that its function might be conserved

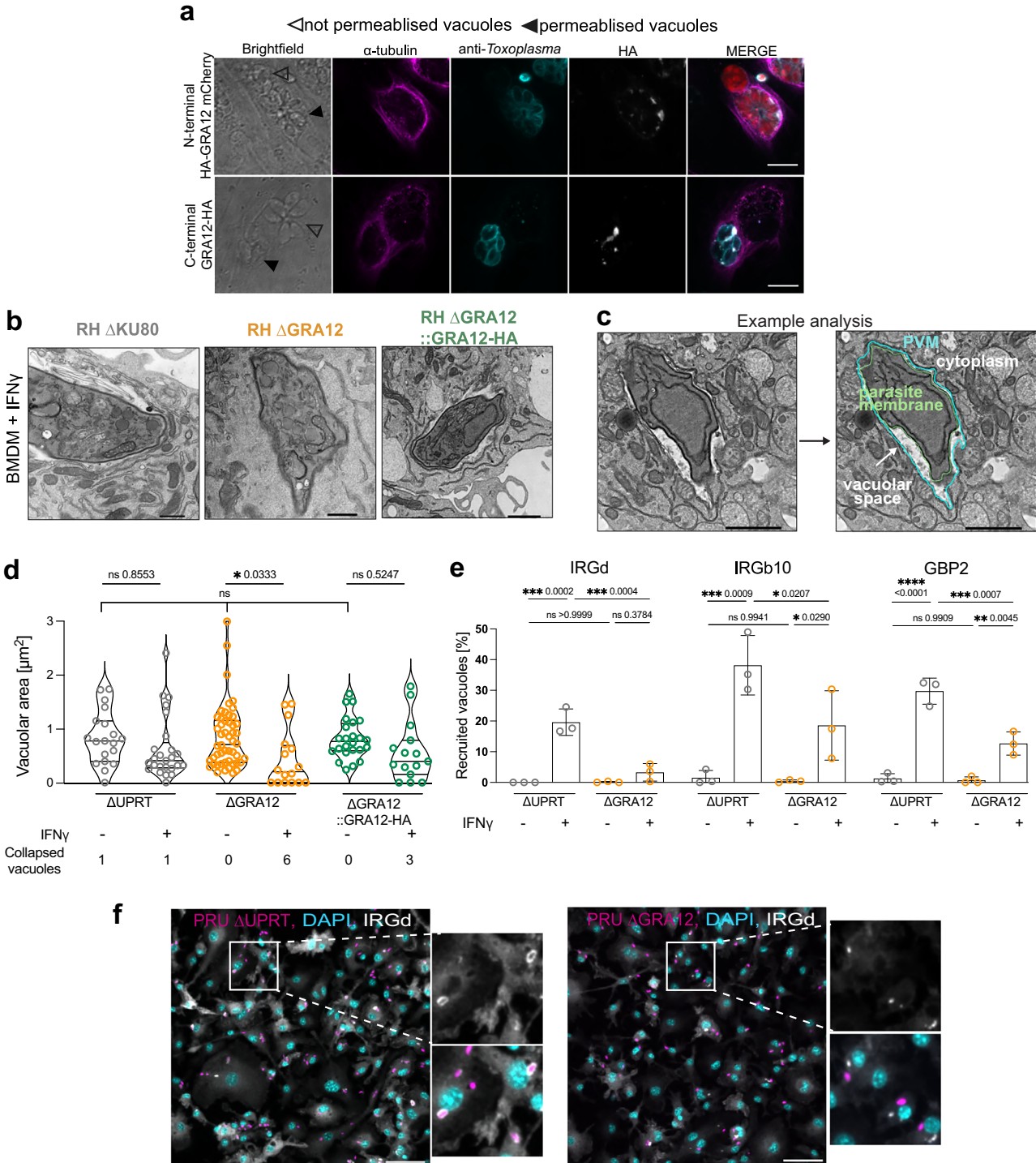

**Fig. 5 | Parasites lacking GRA12 have a collapsed intravacuolar space and are less targeted by host immune proteins. a** Immunofluorescence localisation of GRA12 in partially permeabilised human fibroblasts infected with the N-terminally tagged line (upper panel) or the C-terminally tagged line (lower panel). Filled arrowheads indicate a permeabilised vacuole and empty arrowheads indicate a non-permeabilised vacuole. Scale bar represents 10 μm. **b** Example TEM images of IFNγ-treated PWD/PhJ BMDMs infected with RH ΔKU80, ΔGRA12, ΔGRA12::GRA12-HA for 2 h. Scale bar represents 0.5 μm. **c** Example of analysis for the vacuolar space quantification on a parasite infecting a BMDM. Scale bar represents 1 μm. **d** Quantification of the vacuolar area from the TEM images. Individual vacuoles

were randomly imaged in a single biological experiment. Scatter plot of quantified area. When the PVM was not visible, the vacuolar area was imputed to 0.01 μm². Significance was tested using the Kruskal-Wallis test. **e** Quantification of the recruitment of host proteins (IRGd, IRGb10 and GBP2) to the PVM after 90 min infection of IFNγ-treated or untreated C57BL/6J BMDMs with PRU ΔUPRT or ΔGRA12 parasites. The bar graph shows the median value ± SD, N = 3. Significance was tested using the Tukey's multiple comparisons. **f** Representative images of anti-IRGd stained infections with PRU ΔUPRT (left panel) or PRU ΔGRA12 (right panel). Scale bar represents 50 μm. Source data are provided as a Source Data file.

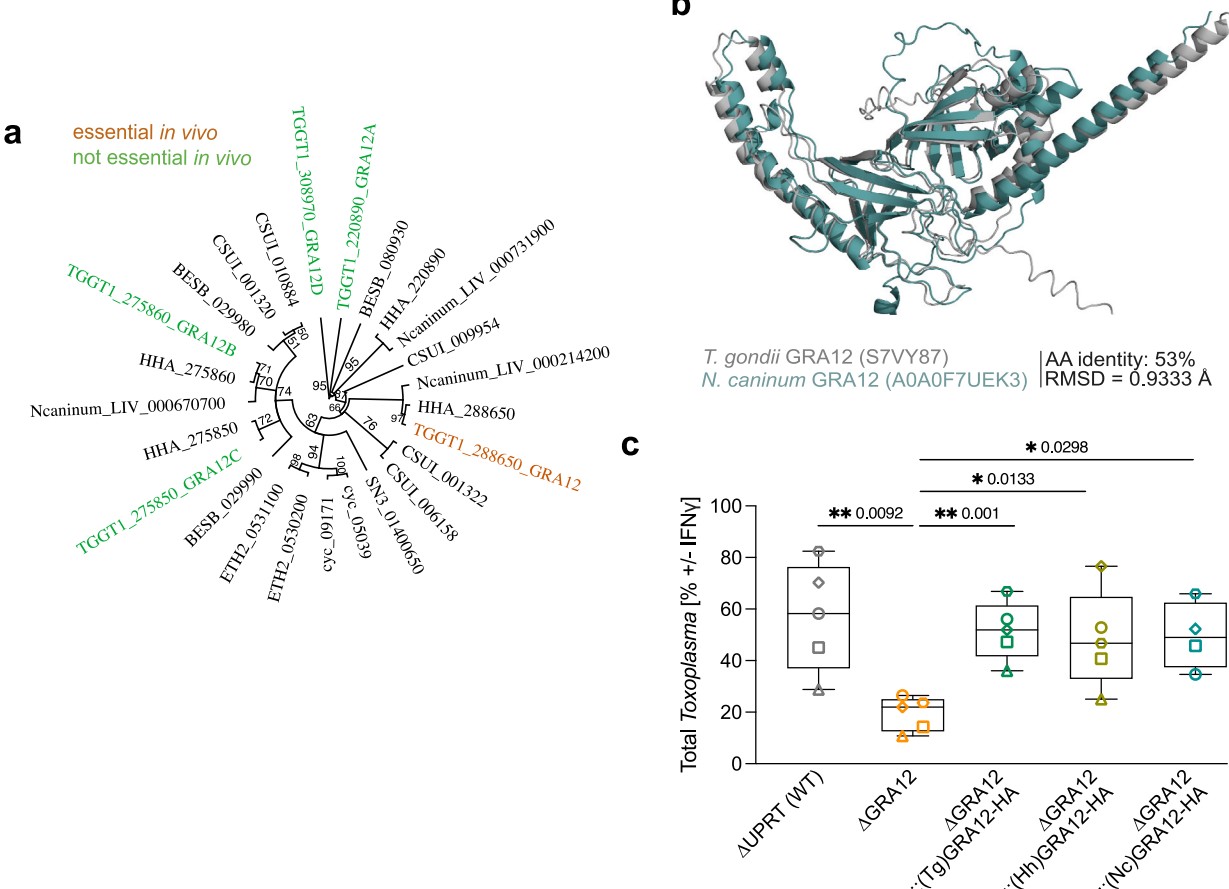

**Fig. 6 | GRA12 is an intravacuolar protein conserved across Coccidia.**
**a** Neighbor-Joining tree of Coccidian parasites GRA12 homologues built with Bootstrap method and considering GRA12D as outgroup. The bootstrap support values, based on 100 replicates, are indicated above each node. **b** Structural overlay of *Toxoplasma* GRA12 (grey) and *N. caninum* GRA12 homologue (teal) as predicted by AlphaFold2. Root Main Square Deviation (RMSD) was calculated using PyMol. **c** Relative *Toxoplasma* growth in IFNγ-treated versus untreated BMDMs. BMDMs were infected with the RH ΔGRA12 strain, or the RHΔGRA12 strain complemented

with *Toxoplasma* (Tg) GRA12 or the *H. hammondi* (Hh) or *N. caninum* (Nc) GRA12 homologues, or the RH ΔUPRT strain as control for 24 h before a plate reader quantification of the mCherry signal as proxy for parasite growth. The box-plot shows the median value ± SD and the whiskers show minimum and maximum values. Significance was tested using the One-way Anova test with the Benjamini, Krieger and Yekutieli FDR correction, $N > = 4$. Source data are provided as a Source Data file.

in these other Apicomplexa. It would be interesting to assess how far the conservation of GRA12 extends by testing whether the most distant GRA12 homologue, which is in Eimeria *spp.* (25% shared amino acid sequence), is still able to complement the *Toxoplasma* GRA12. Type II interferon is also important for clearance of Coccidian parasites, including *N. caninum* and *H. hammondi*[61,62], supporting a similar function of GRA12 to survive the IFNγ-mediated clearance of both parasite species. However, while the murine host is key in the life cycle of *H. hammondi*[62], *N. caninum* only sporadically infects mice in nature[63], suggesting that the function of GRA12 might not be exclusive to murine cells.

In addition to GRA12, we found that deletion of GRA45, GRA61, GRA50 and GRA23 also reduces parasite survival in vivo in all conditions tested here. GRA45 has previously been shown to act as a chaperone to insert proteins into the PVM, for example GRA23, which explains the presence of both proteins as hits[46]. While GRA12 was found to have a partial association with the PV membrane[55,64], for which GRA45 may be required, our data shows that both the N- and C-termini of GRA12 localise within the PV. This suggests that GRA12 does not span the membrane and probably functions independently of GRA45. The function of GRA50 and GRA61 is unknown, and future studies should explore a possible function of these proteins. However, our proteomics data did not identify them as interaction partners of

GRA12, suggesting that they probably function independently of GRA12, or at least not in a complex. Although we may have missed some exported proteins in our study for technical reasons (i.e., not enough datapoints or missing annotation as exported protein at the time of the study[65]), we likely identified the vast majority of virulence factors important in the tested conditions. Our results, together with previous studies that focussed on virulence factors in human cells where GRA12 does not play an important role[44,46,51], suggest that individual virulence factors have evolved to protect the parasite in different species.

We also identified strain-specific effector proteins. Most of these have not been discussed here, but ROP18 behaves as expected, lending support to the robustness of the screens: it is required to confer high virulence to RH and VAND strains in susceptible murine subspecies, is less important for PRU survival and is dispensable in VEG. Surprisingly, the three recently identified secreted members of a protein complex required to prevent parasite IFNγ-mediated clearance in human fibroblasts (GRA57, GRA70, GRA71)[44,51] show a fitness defect in the VAND strain isolated from a lethal human infection, but not in any other strain. This is an important result and warrants further investigation. Positive selection of this protein complex in a zoonotic reservoir could explain the increased virulence of South American *Toxoplasma* strains in humans.

## Method

### Ethics statement

All mouse work was approved by the UK Home Office (project licence P1A20E3F9) and the Francis Crick Institute Ethical Review Panel and carried out in accordance with the UK Animals (Scientific Procedure) Act 1986 and European Union directive 2010/63/EU.

### Animal work

Different strains of male and female *Mus musculus* inbred laboratory mice were used: the C57BL/6J strain, the *M. m. musculus* PWD/PhJ strain and the *M. m. castaneus* CAST/EiJ strain. Non-regulated procedures were performed on Wistar rats (kindly provided by the Biology Research Facility of the Francis Crick Institute) for tissue collection. All animals were bred and housed under pathogen-free conditions in the biological research facility at the Francis Crick Institute.

Bone Marrow-Derived Macrophages (BMDMs) were derived from all murine inbred laboratory strains mentioned above as well as from Wistar rats. Animals were sacrificed by cervical dislocation and the bone marrow collected for extraction of monocytes with a protocol provided by the Wack lab. Bone marrow cells were seeded in 15 cm diameter bacterial Petri dishes (Falcon) and differentiated into BMDMs for 6–7 days with complete RPMI 1640 Medium ATCC modification (Gibco A1049101) supplemented with 10% heat-inactivated FBS (Gibo 10500064), 100 U/ml Penicillin-Streptomycin (Gibco), 50 µM 2-mercaptoethanol (Gibco) and 20% L929 conditioned media. Following differentiation, BMDMs were resuspended in a solution of cold PBS with 2% FCS and 2 mM EDTA (ThermoFisher), and either used for experiments with complete medium without 2-mercaptoethanol or frozen in 9:1 FCS/DMSO until use.

### Cell culture and parasite strains

Primary HFFs and MEFs (ATCC) were maintained in Dulbecco's modified Eagle's high glucose and GlutaMAX™ supplemented medium (DMEM 61965059) with 10% FBS at 37 °C and 5% CO2. *Toxoplasma* strains VAND (gift from Eva Frickel), GT1 and VEG (gift from Martin Blume), RH ΔHXGPRT[66], PRU ΔHXGPRT (gift from Dominique Soldati, as in ref. [66]) and RH ΔKU80[67] were maintained by growth in confluent HFFs and passaged every 2–3 days. Parasites were regularly tested for *Mycoplasma* spp. contamination at the in-house Cell Services Facility. To prepare extracellular parasites for infection, infected HFFs were syringe lysed with a 23 G needle, filtered through a 5 µm sterile filter (Millipore) to remove debris and counted in a haemocytometer chamber prior to dilution to the desired concentration.

### CRISPR-Cas9 screens in vivo and in vitro

CRISPR-Cas9 screens were performed according to previously optimised protocols[42–44] and described as follow. All primers used in this work are listed in Supplementary Data 1.

### Library generation

Screens in RH ΔHXGPRT and PRU ΔHXGPRT strains were performed with a sgRNA library cloned into the previously created vector pCas9-GFP-HXGPRT::sgRNA[68]. For screens in the VAND and VEG strains a novel vector pCas9-GFP-DHFR-TS::sgRNA was created by Gibson cloning the DHFR-TS selection cassette amplified with primers 1–2 from the template GRAx-TGGT1-069070-DHFR-TS in pCas9-GFP-HXGPRT::sgRNA double-digested with SmaI/EcoRI (NEB). PCRs were performed with the proof-reading polymerase KOD (Sigma-Aldrich), and plasmids verified by sequencing with primers 3–7. The pool of ssDNA oligonucleotides encoding the protospacer sequences was selected from an arrayed library using an Echo 550 Acoustic Liquid Handler (Labcyte) in three independent events to minimise imprecisions in the automated liquid handling, and then pooled. The pooled oligonucleotides were integrated in both CRISPR vectors by Gibson cloning after digestion with PacI/NcoI (NEB), resulting in pool libraries of 1240 sgRNAs and 1247 sgRNAs for the HXGPRT and DHFR-TS vectors respectively, and an average of 5 sgRNAs/gene. Individual libraries were sequenced by Illumina sequencing as previously described[42]. Conservation of the target protospacer sequences across all four *Toxoplasma gondii* genomes (ToxoDB, release 68) investigated in this study (ME49 as reference for PRU, GT1, VEG and VAND) was established using Bowtie2[69] and the crisprBowtie package in R[70]. To first control the conserved alignment of protospacers across strains, strict bowtie alignments allowing for 0 mismatches were performed, resulting in 99.61% (ME49), 98.3% (GT1), 98.15% (VEG) and 96.92% (VAND) conservation. Target information for protospacers that did not align to the ME49, VEG or VAND genome were collected and filtered for the number of protospacers that did not align in 3 or more cases. To further investigate effects of potential off-target activity on the number of alignments, mismatch constraints were relaxed to 1–3 mismatches. The list of protospacer sequences, with their mismatch alignments across strains, missing targets (less than 3 guides aligning with 0 or 1 mismatches) and the corresponding summaries are shown in Supplementary Data 2.

### CRISPR pool creation

Parasite transfection was performed in 3–5 replicates and each replicate was sequenced to ensure transfection reproducibility. Libraries were linearised with KpnI-HF or NheI-HF (NEB) for the HXGPRT or DHFR-TS vectors respectively, with an overnight digestion prior to phenol-chloroform purification and transfection with the P3 Primary Cell 4D-Nucleofector kit (Lonza V4XP-3032) in a Amaxa 4D Nucleofector (Lonza AAF-1003X) with the programme EO-115. Integration of the pCas9-GFP-HXGPRT::sgRNA or pCas9-GFP-DHFR-TS::sgRNA libraries was induced upon treatment with 25 µg/ml Mycophenolic acid (Sigma-Aldrich) and 50 µg/ml Xanthine (Sigma-Aldrich) the following day, or with 2 µM Pyrimethamine (Merck) 2 h post transfection respectively. Three days post transfection, parasites were syringe lysed and added to fresh HFFs monolayers with 100 U/ml Benzonase (Merck) overnight to remove traces of input DNA. Six to eight days post transfection, samples were taken for gDNA preparation or used as in vivo/in vitro inoculum. To establish the transfection efficiency, parasites after transfection with and without the selection drug were used to infect HFFs in a plaque assay and fixed after 7–10 days. The transfection efficiency expressed in percentage was calculated with the relative number of surviving parasites following selection, normalised to the not transfected parasite population.

### In vivo selection

Different combinations of parasite and mouse strains were used as specified throughout the manuscript. The inoculum was diluted at 1E6 parasites/ml in PBS and mice were injected intraperitoneally with 200 µl equal to 2E5 parasites for all but VEG, for which a dose of 5E5 parasites was used to account for higher restriction of this parasite strain in vivo. Eight- to sixteen-week-old male mice were injected, and each in vivo screen was performed on five mice. Five days post infection, parasites and peritoneal cells were isolated by double peritoneal lavage with 5 ml of PBS. Peritoneal cells were pelleted at 500 g for 5 min, resuspended in PBS and intracellular parasites were syringe lysed with 27 G and 30 G needles and used to infect HFFs in DMEM supplemented with 100 U/ml Pen/Strep. After one lytic cycle, expanded parasites were pelleted for gDNA extraction.

### In vitro selection

BMDMs from PWD/PhJ mice were used for the RH ΔHXGPRT screen to match the mouse strain-parasite strain combination used in the in vivo screen. 7E6 BMDMs were seeded and treated for 24 h with 10 U/ml murine IFNγ (Gibco PMC4031) or left untreated prior to infection. To limit co-infections, 1.4E6 parasites per dish corresponding to a MOI 0.2 and a coverage of 1230 parasites/sgRNA were used. At 48 hpi BMDMs

were scraped, syringe lysed with a combination of 23 G, 27 G and 30 G needles, and parasites from round 1 were used at a MOI 0.2–0.3 for infection of round 2 in BMDMs similarly prepared. Each condition was performed in triplicate. Similarly to the in vivo screen, parasites were expanded for one lytic cycle in fibroblasts, before gDNA extraction and sgRNA amplification for sequencing.

### Illumina sequencing of sgRNA and data analysis

Genomic DNA was extracted from samples using DNEasy Blood kit (Qiagen), then guide sequences were amplified by nested PCR using KAPA HIFI Hotstart PCR kit (Kapa Biosystems KK2501) and KAPA Pure beads (Kapa Biosystems KK8000). Primers 8–27 were used for nested PCRs. Purified PCR products were then sequenced on a HiSeq400 (Illumina) with paired end 100 bp reads at a minimum read depth of 5 million reads/sample. gRNA sequences were aligned to the reference library and counts were normalised using the median of ratios, then sgRNA not present in all samples or in less than three peritoneal samples were removed from the analysis. For the in vivo CRISPR screens, the median L2FC for each sgRNA was calculated from the normalised counts in the inoculum sample relatively to the plasmid (Median L2FC in vitro), or from the peritoneal samples at day five compared to the inoculum (Median L2FC in vivo). For the in vitro CRISPR screens, the median L2FC for each sgRNA was calculated from the normalised counts in the unstimulated BMDMs relative to the inoculum (Median L2FC BMDM -IFNγ vs HFF), or from the IFNγ-stimulated BMDMs relative to unstimulated BMDMs after the second round of selection (Median L2FC BMDM + vs - IFNγ). The MAD score across gRNA L2FCs targeting each gene was calculated, and genes with the highest 1.5% of MAD scores were removed from the analysis. A DISCO score based on the local FDR-corrected q-value was calculated for each L2FC.

### Creation and validation of the VAND mutant strains

To create VAND mutant strains, the ProGRA1::mCherry::T2A::HXGPRT::TerGRA2 plasmid[42] encoding the repair template mCherry::T2A::HXGPRT was modified by replacing *Hxgprt* with the Chloramphenicol acetyltransferase (CAT) sequence from the pG140::DiCre plasmid[71], amplified with primers 28–31 and Gibson cloned. The resulting ProGRA1::mCherry::T2A::CAT::TerGRA2 plasmid was validated with primers 3 and 32. VAND ΔGRA12, ΔROP18 and ΔUPRT strains were created by co-transfecting VAND with the protospacer-encoding pCas9::GRA12, pCas9::ROP18 and pCas9::UPRT plasmids (created with primers 33–35 or with the original pCas9::UPRT plasmid, and validated by sequencing with primer 36) and the relative repair templates amplified with primers 37–42 from the Pro-GRA1::mCherry::T2A::CAT::TerGRA2 plasmid. The disruption of the endogenous locus was PCR validated with primers 43-48 and by Nanopore sequencing as follow. Three libraries were prepared from 200 ng of genomic DNA per sample using the Nanopore SQK-RBK114 kit. Sequencing was carried out on a PromethION flow cell (R.10.4.1) using a P2 Solo device for 24 h. Basecalling was performed with Dorado v0.7.3 (Dorado), utilising the "sup" model, and the reads were aligned to the ToxoDB-65_TgondiVAND genome assembly reference from ToxoDB release 49 (VAND). The aligned reads were indexed and sorted for future visualisation. Post-demultiplexing, Dorado was used again to detect and trim primers and adapters from the reads. Mapped reads were visualised against the reference genome using IGV v2.16.2[72]. Nanopore sequence reads alignment showing disruption of the endogenous loci in VAND ΔGRA12, ΔROP18 and ΔUPRT strains, and integration of the repair template is represented in Supplementary Fig. 10.

To create the complemented strain VAND ΔGRA12::GRA12-HA, the 5'UTR and cDNA of GRA12 (TGVAND_288650) were Gibson cloned in the pUPRT vector[43] with primers 49–54, and the plasmid pUPRT::-GRA12 was verified by sequencing with primers 41, 49–50. VAND ΔGRA12 parasites were co-transfected with the ScaI (NEB) digested

pUPRT::GRA12 plasmid and the pCas9::UPRT plasmid, and selected for integration in the *Uprt* locus with 5 μM 5'-fluo-2'-deoxyuridine (FUDR, Sigma F0503) the day after transfection. A clonal population of the resulting VAND ΔGRA12::GRA12-HA parasites was validated by PCR with primers 57–60, by western blot and immunofluorescence detection of the protein. PCRs for cloning were performed with the proof-reading polymerase ClonAmp HiFi PCR Premix (Takara) and PCR for validation of strains were performed with EmeraldAmp GT Master Mix (Takara).

### In vivo infections

PWD/PhJ mice 8–12 weeks old were infected intraperitoneally with different doses of parasites in 200 μl PBS. Immediately after infection, fitness of the parasites used as inoculum was assessed via plaque assay to establish the precise parasite dose. Mice were monitored and weighed regularly for the duration of the experiments or until reach of the humane endpoint. To determine the number of cysts in the brain of infected animals, mice were euthanised at 28 days post infection. The brain was homogenised in 1 ml PBS and stained with Fluorescein-conjugated Dolichos Biflorus agglutinin (1:200, Vector Laboratories RL-1031) for 1 h at room temperature. Fluorescently labelled cysts were counted using a Ti-E Nikon microscope with a 40x magnification in a 1:1 dilution of homogenate and PBS. At least one third of the brain homogenate was analysed for cysts presence.

### Plaque assays

HFFs and MEFs were grown to confluency in T25 flasks and infected with 100–200 parasites to grow undisturbed for 10 days for RH derived strains and 14 days for other strains. Up to 25,000 parasites were plaqued to determine the transfection efficiency of the CRISPR transfections. Cells were fixed and stained in a solution with 0.5% w/v crystal violet (Sigma), 0.9% w/v ammonium oxalate (Sigma), 20% v/v methanol in distilled water then washed with tap water. Plaques were imaged on a ChemiDoc imaging system (BioRad) and measured in Fiji[73].

### Creation of the RH ΔKU80-derived strains

To create the RH ΔKU80 GRA12 strain, RH ΔKU80 parasites were co-transfected with the pCas9::GRA12 plasmid and the repair template amplified from the ProGRA1::mCherry::T2A::HXGPRT::TerGRA2 plasmid with primers 37–38, and selected with M/X 24 h after transfection. A clonal population of RH ΔGRA12 strain was validated by PCR with primers 43–44, 61–62. The GRA12 C-terminal endogenously tagged RH GRA12-HA strain was created by co-transfection of RH ΔKU80 parasites with the pCas9::GRA12-CT plasmid encoding a protospacer targeting the 3' UTR (created with primers 33 and 63, and validated with primer 36) and the repair template amplified with primers 64–65 from the HA-TerGRA2::ProDHFR-HXGPRT-TerDHFR plasmid[43], and selected with M/X 24 h after transfection. A clonal population of the resulting RH GRA12-HA strain was validated by PCR with primers 57–58, 60 and 66, by western blot and by immunofluorescence detection of the protein.

The GRA12 N-terminally tagged RH ΔGRA12::HA-GRA12 strain and the complemented RH ΔGRA12::GRA12-HA strains with either the *Toxoplasma* (RH ΔGRA12::(Tg)GRA12-HA), *Hammondia hammondi* (RH ΔGRA12::(Hh)GRA12-HA) or *Neospora caninum* (RH ΔGRA12::(Nc) GRA12-HA) homologues were all created by cloning into the *Uprt* locus via co-transfection of pCas9::UPRT and a pUPRT repair plasmid. The plasmid pUPRT::GRA12(GT1) to complement the type I TGGT1_288650 HA-tagged isoform of GRA12 in RH ΔGRA12 and create the RH ΔGRA12::(Tg)GRA12-HA strain, or simply called RH ΔGRA12::GRA12-HA, was created similarly to pUPRT::GRA12 with plasmids 52-54 and 67. In the pUPRT::HA-GRA12 plasmid for N terminal GRA12 tagging, a HA-tag followed by a Gly-Gly linker were inserted at position 47 of the amino acid sequence, between the signal peptide sequence (AA 1-38) and the putative transmembrane domain (AA 93–115) as predicted by Michelin

et al.[55]. The plasmid pUPRT::HA-GRA12 was created similarly to pUPRT::GRA12 with primers 50, 67–71. The *H. hammondi* and *N. caninum* cDNA isoforms were PCR amplified from a codon optimised gBlock (IDT, 72 in Supplementary Data 1) and Gibson cloned to create plasmids pUPRT::(Hh)GRA12 and pUPRT::(Nc)GRA12 with primers 73–80. All complemented plasmid constructs were validated by sequencing with primers 55–56, ScaI-linearised and co-transfected with pCas9::UPRT, and parasites selected for integration by FUDR treatment the day after. All complemented strains were validated by PCR with primers 55–56 and by immunofluorescence detection of the protein. PCRs for cloning were performed with the proof-reading polymerase ClonAmp HiFi PCR Premix (Takara) and PCR for validation of strains were performed with EmeraldAmp GT Master Mix (Takara).

### Immunofluorescence assay and live-cell imaging
HFFs were seeded on 8-well µ-slides (Ibidi 80806) and infected for 24 h, and BMDMs were seeded on 96-well plates (Revvity PhenoPlate 6005430) and infected for 90 min, before fixation in 4% paraformaldehyde (PFA) for 15 min. Samples were then permeabilised in 0.1% saponin for 10 min and blocked with 3% bovine serum albumin (BSA) in PBS for 1 h. For the localisation of the N and C-terminal extremities of GRA12 in Fig. 5a, samples have been treated with 0.001% saponin for 15 s to permeabilise only some but not all PVMs, or fully permeabilised for 10 min or left untreated as controls. All antibodies were incubated in 3% BSA in PBS for 2 h at RT or at 4 °C o/n, followed by 3x washes in PBS and a secondary staining combined with DAPI. Images were acquired on a VisiTech instant SIM (VT-iSIM) microscope using a 150x oil-immersion objective with 1.5 µm z axis steps (Fig. 2a, Supplementary Fig. 3e, Supplementary Fig. 6b and Supplementary Fig. 9c), on the 3i Marianas II LightSheet microscope using a 40x or 60x objectives with 8 µm z axis steps (Figs. 4d, e and 5a, f, Supplementary Figs. 5d, 6a, c and 8a) or on the Nikon Ti-E inverted widefield microscope with a Nikon CFI APO TIRF 100x/1.49 objective and Hamamatsu C11440 ORCA Flash 4.0 camera running NIS Elements (Nikon) (Supplementary Fig. 2b). Resultant stacks were deconvoluted and processed using Microvolution plugin in Fiji. Primary antibodies used were 1:500 rat anti-HA (Roche 11867423001), 1:1000 rabbit anti-*T*oxoplasma (Abcam ab138698), 1:1000 mouse anti-α tubulin (Abcam ab7291), 1:1000 mouse anti-GRA2 and 1:1000 rabbit anti-GRA3 (both gifts from Jean François Dubremetz), 1:1000 rat anti-GRA12 (gift from Maryse Lebrun), 1:1000 rabbit anti-IRGd6 (081/3) and 1:6000 anti-IRGb10 (940/6, both gifts from Jonathan Howard), 1:500 rabbit anti-GBP2 (Proteintech 11854-1-AP), 1:200 mouse anti-Ubiquitin (FK2, Merck ST1200). All secondary antibodies used were from Thermo Fisher and were 1:500 anti-rat AlexaFluor 647 (A21247), 1:500 anti-mouse AlexaFluor 647 (A21235), 1:500 anti-rabbit AlexaFluor 647 (A21244), 1:2,000 anti-rabbit AlexaFluor 488 (A11008), 1:2,000 anti-mouse AlexaFluor 488 (A11029), 1:2,000 anti-rat AlexaFluor 488 (A11006), 1:2,000 anti-rabbit AlexaFluor 405 (A-31556).

For the Live Cell Imaging in Fig. 4d and Supplementary Video 1, C57BL/6J BMDMs were seeded in 8-well µ-slides (Ibidi 80806), pre-treated for 24 h with 100 U/ml mIFNγ or left untreated, and infected with RH ΔUPRT, ΔGRA12 or ΔGRA12::GRA12-HA at a MOI of 1. After 30 min, cells were washed twice and imaged for 12 h with 5 min interval on a 3i Marianas II LightSheet microscope using a 40x objective.

### Western blot
Syringe-lysed parasites from HFFs culture were used for western blots for strain validation, while 1E6 C57BL/6J BMDMs were seeded in 6-well plates to identify cell death pathways by western blots. BMDMs were treated for 24 h with 100 U/ml mIFNγ or left untreated as control, and infected for 8 h with RH ΔUPRT, ΔGRA12, ΔGRA12::GRA12-HA at a MOI 3. BMDMs were treated with 0.25 mM H$_2$O$_2$ for 4 h, or with 0.5 µM Staurosporine (Cayman 81590) for 8 h or with 20 µM Z-VAD (ApexBio A1902) for 30 min before 7 h treatment with 20 ng/ml TNF-α

(Biolegend 718004) and 100 nM SM-164 (Cayman 28632), as positive control for death by necrosis, apoptosis and necroptosis respectively. Proteins from the supernatant were precipitated with 4x ice-cold acetone for 1 h in −20 °C, then were spun down at 13,000 g for 10 min and the pellet let dry and resuspended in 4x sample buffer solution (Bioworld BW-21420018-3) supplemented with 2-mercaptoethanol. BMDMs were washed once with cold PBS and lysed on ice in RIPA buffer (Thermo Fisher 89901) supplemented with 2× cOmplete Mini EDTA-free Protease Inhibitor Cocktail (Roche) for 30 min, then spun at 13,000 g 5 min to remove the insoluble fraction. Samples were added with 4x sample buffer solution supplemented with 2-mercaptoethanol and boiled for 5 min at 95 °C together with the supernatants. Samples were separated on a SDS-PAGE on precast 4–15% TGX stain-free gels (Bio-Rad) and the Precision Plus Protein All Blue ladder (Bio-Rad) was used. Proteins were transferred to a nitrocellulose membrane for 7 min using the Turbo midi protocol on the Trans-Blot Turbo transfer system (Bio-Rad). Membranes were blocked in blocking solution (5% milk in PBS with 0.1% Tween-20) for 1 h RT or o/n at 4 °C, followed by incubation with primary antibodies in blocking solution either for 2 h RT or o/n at 4 °C, and of secondary antibodies for 1 h RT. Primary antibodies used were 1:1000 rabbit anti-*T*oxoplasma (Abcam ab138698), 1:10,000 rabbit anti-ROP18 (gift from David Sibley), 1:1000 rat anti-HA Horseradish Peroxidase (HRP) conjugated (Roche 12013819001), 1:1000 rabbit anti-caspase 8 (Cell Signalling Technology 4927), 1:1000 rabbit anti-cleaved caspase 8 (Cell Signalling Technology 8592), 1:1000 rabbit anti-MLKL(phospho Ser345, Abcam ab196436), 1:10,000 mouse anti-β actin (Sigma A2228) and 1:1000 rabbit anti-HMGB1 (Abcam ab18256). Secondary antibodies used were 1:10,000 goat anti-rabbit HRP (Insight Biotechnology 474–1506) and 1:10,000 goat anti-mouse HRP (Insight Biotechnologies, 474–1806). HRP was detected using the Immobilon Western Chemiluminescent HRP substrate (Millipore), visualised on a ChemiDoc imaging system (BioRad) or a GE Amersham Imager 680.

## Restriction assays
### High-content imaging
Following previously optimised protocols[43,44], MEFs or 75,000 BMDMs were seeded in black clear-bottom 96-well µ-Plate imaging plates (Ibidi 89626). Cells were pre-stimulated for 24 h with IFNγ according to host species: 100 U/ml mIFNγ (Thermo Fisher, Gibco PMC4031) for murine BMDM and MEFs, and 25 U/ml mIFNγ for rat BMDM. Cells were infected at a MOI of 0.3 for 24 h, then fixed in 4% PFA and stained with 5 µg/ml DAPI and 5 µg/ml CellMask Deep Red (Invitrogen C10046). Plates were imaged using the Opera Phenix high-content screening system, with 25 images and 5 focal planes acquired per well, and an automated analysis of infection phenotypes was performed using Harmony v5 (Revvity) as previously described[44]. Data is reported as the mean proportion of each factor (vacuole number and total *Toxoplasma* number) in IFNγ-treated wells relative to untreated wells. Differences between strains were tested by paired two-sided *t* test with the Benjamini, Krieger and Yekutieli FDR correction.

### Cytation5 plate reader
Cells were prepared, infected and analysed following the protocol above. As previously performed[44], plates were imaged on a Cytation5 plate reader (BioTek) using a 20x objective in a 2 × 2 tile. Since all strains express an mCherry reporter, the total signal was measured with the Texas Red filter (Ex/Em 586/647) and used as proxy of *Toxoplasma* growth.

### Propidium uptake assay
Black clear-bottom 96-well plates were seeded with 75,000 BMDMs per well and stimulated with 100 U/ml mIFNγ for 24 h. Cells were infected at an MOI of 3 in a media solution with 5 µg/mL propidium iodide (Thermo Fisher P3566). Images of each well were acquired every

30 minutes until 14 h post-infection in two different systems due to the lab relocation in a different institute. Similarly to previously established protocols[43], experiments in Fig. 4a, b and Supplementary Fig. 4a, b, were analysed on a Nikon Ti-E inverted widefield fluorescence microscope with a Nikon CFI Plan Fluor 4x/0.13 objective and Hamamatsu C11440 ORCA Flash 4.0 camera running NIS Elements (Nikon). At the end of the experiment, 10% Triton X-100 solution was added to a final concentration of 1% v/v to fully permeabilise the cells and a final image was captured of each well. In each image, the total fluorescence signal was measured. The percentage of propidium iodide uptake in each well at each timepoint was calculated by subtracting the first measurement at 1 hpi to remove background fluorescence signal and normalising the total fluorescence intensity following full permeabilisation to 100% uptake. Experiments in Fig. 4f, g and Supplementary Fig. 4f were analysed in an Agilent Cytation C5 plate reader with a 4x/0.13 objective and the Texas Red filter (Ex/Em 586/647). 30 min after infection at a MOI of 5, cells were washed with media with propidium iodide, with or without IFNγ based on the initial treatment, and with 1 µM ML10 (gift of Michael Blackman), or left untreated as control. All acquired images were analysed with the built-in software, and host cell nuclei positive to propidium iodide were automatically counted. Following Triton X-100 permeabilisation, another round of imaging was performed, and the number of dead cells was expressed as percentage of propidium iodide positive cells at each time point and condition out of the total in the final read. In both settings, imaging was performed at 37 °C and with 5% CO_2. All conditions were performed in technical triplicate, of which the mean was taken to represent each biological replicate. Differences between strains were tested at 9 hpi by One-Way ANOVA with paired two-sided $t$-test with the Benjamini, Krieger and Yekutieli FDR correction.

### Transmission electron microscopy
1E5 PWD/PhJ BMDM were seeded on 10 mm Poly-D-Lysine (Gibco A3890401) treated glass coverslips, treated for 24 h with 100 U/ml mIFNγ or left untreated, and infected with 5E5 parasites of the RH ΔKU80, ΔGRA12, ΔGRA12::GRA12-HA strains for 2 h. Cells were then washed 3x with PBS and fixed in 2.5% glutaraldehyde and 4% formaldehyde in 0.1 M phosphate buffer (PB; pH 7.4) for 30 min at room temperature. Samples were washed 2x with 0.1 M PB and stained with 1% (v/v) osmium tetroxide (Taab)/ 1.5% potassium ferricyanide (Sigma) for 1 h at 4 °C. All remaining processing steps were performed as per previously established protocols[44] using the Pelco BioWave Pro+ microwave (Ted Pella) with a Steady Temp set to 21 °C. In brief, samples were incubated in 1% (w/v) tannic acid in 0.05 M PB (pH 7.4; Sigma) for 14 min under vacuum in 2 min cycles alternating with/without 100 W power, followed by 1% sodium sulphate in 0.05 M PB (pH 7.4; Sigma) for 1 min without vacuum at 100 W. Samples were then washed in dH20 and dehydrated through an ethanol series (70%, 90% and 100%), followed by 3x incubations in 100% acetone at 250 W for 40 s without vacuum. Samples were exchanged into 1:1 acetone: Epon resin (Taab) overnight, then into 100% resin for 6 h before polymerising at 60 °C for 48 h. The glass coverslip was removed with liquid nitrogen. Samples were sectioned using a UC7 ultramicrotome (Leica Microsystems) and 70 nm sections were picked up on Formvar-coated 2 mm copper slot grids (Gilder Grids). All grids were post stained with 3% lead citrate for 2 min. For all six conditions, sections containing *Toxoplasma gondii* in a horizontal orientation were viewed at 5000x, 6000x, 8000x, 10000x and 15000x using a 120 kV 1400FLASH TEM (JEOL), and images were acquired with a JEOL Matataki Flash sCMOS camera. The vacuolar area was quantified in Fiji[73] measuring the difference in area between the parasite membrane and the parasitophorous vacuole membrane (PVM). When the PVM was not visible, the vacuolar area was imputed to 0.01 µm².

### Griess assay for nitric oxide quantification
75,000 C57BL/6J BMDMs were seeded in a 96 well plate and treated for 24 h with 100 U/ml IFNγ and 0.2 µg/ml LPS (InvivoGen 0111:B4), or left untreated as control before infection with RH ΔUPRT, ΔGRA12, ΔGRA12::GRA12-HA in a ratio cell:parasite of 1:5 and a total volume of 100 µl phenol-free media to not interfere with the colorimetric reaction. Uninfected cells were used as control and all conditions were performed in technical triplicate. After 24 h the plate was spun 1000 rpm for 1 min and 50 µl of supernatant were mixed in a 96 well plate in a 1:1 ratio with Griess reagent freshly prepared (1:1 solution of 0.1% N-(1-Naphthyl) ethylenediamine dihydrochloride (Sigma-Aldrich) in distilled water and 1% sulfanilamide (Sigma-Aldrich) in 5% phosphoric acid (Sigma-Aldrich)). A solution of sodium nitrite 0–100 µM was used to calibrate the standard curve. The plate was briefly shaken and incubated at room temperature for 5 min, then read at 540 nm in a Biotek Sinergy H1 Neo2 machine. To confirm the activity of the iNOS inhibitor N-(3-(Aminomethyl)benzyl)acetamidine (1400W-HCl, Selleck Chemicals S8337), 100 µM of the same were added 1 h before 0.2 µg/ml LPS stimulation of C57BL/6J BMDMs pretreated with 100 U/ml IFNγ or left untreated as control.

### Immunoprecipitation and mass spectrometry
20E6 PWD/PhJ BMDMs were seeded in 15 cm Petri dishes and treated for 24 h with 10 U/ml mIFNγ before infection with RH ΔKU80 or RH GRA12-HA in triplicate for 24 h at a MOI 0.3. Infected cells were washed 2x in cold PBS then lysed in cold immunoprecipitation (IP) buffer (10 mM Tris, 150 mM NaCl, 0.5 mM EDTA and 0.4% NP40, pH 7.5 in H2O, supplemented with 2x cOmplete Mini EDTA-free Protease Inhibitor Cocktail). Lysates were syringe-lysed 6x through a 30 G needle and left on ice for 1 h, then centrifuged at 2000 $g$ for 20 min to remove the insoluble fraction. Soluble fractions were added to 30 µl/sample anti-HA agarose beads (Thermo 26182), then incubated o/n at 4 °C with rotation. Beads were washed 3x with cold IP buffer for 10 min each, then proteins were eluted in 30 µl 4x Sample Loading Buffer supplemented with DTT and boiled for 5 min at 95 °C.

Approximately 20 µl of each IP elution was loaded on a 10% Bis-Tris gel and run into the gel for 1 cm, then stained with InstantBlue Coomassie Protein Stain (Abcam ab119211). Proteins were alkylated in-gel prior to digestion with 100 ng trypsin (modified sequencing grade, Promega) overnight at 37 °C. Supernatants were dried in a vacuum centrifuge and resuspended in 0.1% trifluoroacetic acid (TFA), and 1 to 10 µl of acidified protein digest was loaded onto a 20 mm × 75 µm Pepmap C18 trap column (Thermo Scientific) on an Ultimate 3000 nanoRSLC HPLC (Thermo Scientific) prior to elution via a 50 cm × 75 µm EasySpray C18 column into a Lumos Tribrid Orbitrap mass spectrometer (Thermo Scientific). A 70' gradient of 6% to 40% B was used to elute bound peptides followed by washing and re-equilibration (A = 0.1% formic acid, 5% DMSO; B = 80% ACN, 5% DMSO, 0.1% formic acid). The Orbitrap was operated in "Data Dependent Acquisition" mode followed by MS/MS in "TopS" mode using the vendor supplied "universal method" with default parameters.

Raw files were processed to identify tryptic peptides using Maxquant (maxquant.org) and searched against the *Toxoplasma* (ToxoDB-56_TgondiiGT1_AnnotatedProteins) and Murine (Uniprot, UP000000589) reference proteome databases and a common contaminants database. A decoy database of reversed sequences was used to filter false positives, at peptide and protein false detection rates (FDRs) of 1%. $T$ test-based volcano plots of fold changes were generated in Perseus (maxquant.net/perseus) with significantly different changes in protein abundance determined by a permutation-based FDR of 0.05% to address multiple hypothesis testing. Raw data are provided in Supplementary Data 8.

## Phylogenetic analysis of GRA12-like proteins and protein structure prediction

Amino acid sequences of homologues and paralogues of GRA12 within Apicomplexa were obtained from the ToxoDB database[74]. A global alignment of the 25 sequences was performed in Geneious Prime (version 2024.0.2. Cost matrix Blosum62, gap open penalty 12, gap extension penalty 3 and refinement iterations 2). A Neighbor-Joining Tree was built with Bootstrap method with 100 replicates and imputing GRA12D (TGGT1_308970) as outgroup.

GRA12's structures from *Toxoplasma* (UniProt S7VY87) and *Neospora* (A0A0F7UEK3) as predicted by the AlphaFold protein structure database (version 3)[75], were overlapped on PyMOL (version 2.4.1), and the Root Mean Square Deviation (RMSD) was calculated to infer protein structure similarity. The C and N termini were not included in Fig. 6b.

## Statistics and reproducibility

Data was analysed in GraphPad Prism v10 and details of the analyses are reported in each legend. No statistical method was used to predetermine sample size, and no data were excluded from the analyses. The qualitative analysis of the TEM experiment was performed blinded by three investigators. Microscopy images in Supplementary Fig 6a, c were analysed blinded. The other experiments in the manuscript were not blinded. All experiments with fluorescent microscopy and western blots were performed with a minimum of two biological repeats. Other experiments in the manuscript counted a minimum of three biological repeats. Statistical significance was specified throughout the manuscript and set as: ns – not statistically significant, $*p < 0.05$; $**p < 0.01$; $***p < 0.001$, $****p < 0.0001$.

## Reporting summary

Further information on research design is available in the Nature Portfolio Reporting Summary linked to this article.

## Data availability

All data generated in this study are provided in the Supplementary Information and Source Data files. Source data are provided with this paper.

## Code availability

The code to reproduce the analyses in this manuscript is available on Github[76].

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

## Acknowledgements

We thank the Science Technology Platforms at the Francis Crick Institute – and in particular the Biology Research Facility, the Advanced Sequencing Facility, the Light Microscopy Facility, the Proteomics Facility and the Cell Services Facility – for their support and help. We thank the Advanced Imaging Facility and Genomics Facility at the Instituto Gulbenkian de Ciência for their support and help. We thank Eva Frickel for the VAND strain and Martin Blume for the VEG and GT1 strains. We thank James Turner for the PWD/PhJ and CAST/EiJ strains, and Sangrithi Mahesh for CAST/EiJ tissue samples. We thank Andreas Wack for providing the BMDMs differentiation protocol, Jonathan Howard, Maryse Lebrun, David Sibley and Jean François Dubremetz for providing antibodies, Michael Blackman for providing reagents and Miguel Soares for providing the L929 cell line. We thank VEuPathDB for providing access to the *Toxoplasma* databases[74]. We thank all members of the Treeck lab, Maximiliano Gutierrez, Beren Aylan and Alice Balard for critical input and reading of this manuscript. This work was supported by an award to M.T. from the Wellcome Trust (223192/Z/21/Z), the FCT - Fundação para a Ciência e a Tecnologia (2023.06167.CEECIND), and by the Francis Crick Institute which receives its core funding from Cancer Research UK, the UK Medical Research Council and the Wellcome Trust (CC2132 and CR2023/030/2123) supporting The Science Technology Platforms (CC0199). F.H. is supported by the Wenner-Gren-Foundations (WGF2024-0027) and F.T. is supported by the Deutsche Forschungsgemeinschaft (TO 1349/1-1). A.N.M. is supported by the FCT (UI/BD/154200/2022).

## Author contributions

Conceptualisation, F.T. and M.T.; methodology, F.T., S.B., and E.L.; investigation, F.T., S.B., E.L., A.N.M., O.S., and J.P.F.; formal analysis, F.T., S.B., and F.H.; visualisation, F.T.; writing – original draft, F.T. and M.T.; writing – review and editing, all authors; supervision, M.T.; funding acquisition, F.T., F.H., and M.T.

## Funding

## Competing interests

The authors declare no competing interests.
