## [Transparent Peer Review file · Nature Communications]

GRA12 is a common virulence factor across *Toxoplasma gondii* strains and mouse subspecies

Corresponding Author: Dr Moritz Treeck

Version 0:

Reviewer comments:

Reviewer #1

(Remarks to the Author)

In this manuscript by Torelli et al., the authors propose that GRA12 is a fundamental virulence gene independent of parasite strain or mouse subspecies. Most studies in *Toxoplasma* so far have been conducted in lab-adapted strains in parasites (e.g., RH and Pru) and mice (e.g., C57BL/6 and BALB/c). However, this does not necessarily represent real-world host-parasite interactions. The authors used a highly virulent atypical strain VAND, and mouse subspecies that possess polymorphisms in IRGs. The authors conducted multiple in vivo CRISPR screens and elegantly showed that ROP18 depends on parasite strains and mouse subspecies as expected. Interestingly, GRA12 is the top hit on any screens. Deletion of GRA12 results in host cell necrosis in unknown mechanisms but is partially dependent on the early egress of parasites. Although the authors do not fully address the function of GRA12, this paper should be of great interest to the field of *Toxoplasma* parasitology.

Major concerns

1. The authors previously designed a gRNA library based on the type II ME49 genome. They aligned gRNA sequences to the type I GT1 genome to validate that the gRNA library can be used for both strains (Young et al., Nat Commun, 2019, PMID: 31481656). Did the authors validate the gRNA library for the type III and VAND genomes? This validation data must be shown in the supplemental data in the way of Young et al. (Nat Commun, 2019, PMID: 31481656).
2. This reviewer needs clarification on Figure S5F. Are these samples treated or non-treated by IFN- γ , or just mislabeled?
3. Line 408-412, the authors described that GRA12 paralogues (GRA12A-D) are not essential for parasite infection in vivo based on the results of published CRISPR screens. However, there is a study that investigates GRA12 paralogues (Guevara et al., mSphere, 2021, PMID: 33883265). This paper showed that GRA12 paralogues are required for acute virulence of low-dose infection and cyst burden. Although this reviewer agrees that GRA12 is the most important for acute infection, this paragraph could be revised, and Guevara et al. should be cited.
4. In Figure 2B, a single band of GRA12-HA is observed around 40 kDa. In Figure S3C, there is another band of GRA12-HA below 25 kDa in RH Δ GRA12::GRA12-HA. Is there also a smaller band in VAND Δ GRA12::GRA12-HA (Figure 2A)? What is the smaller band of GRA12-HA?

Minor comments

1. In Figure S1A, how did the authors calculate the transfection efficiency of each strain? I assume these are GFP-positive rates of parasites, but the details should be described in the material and methods.

Reviewer #2

(Remarks to the Author)

Toxoplasma gondii strains are grouped into lineages (I, II, III, and atypical) that have different levels of virulence in mice and between mouse strains. Taking a novel approach, the authors have designed CRISPR screens to identify virulence factors that are conserved across parasite strains. A strength of the paper is that the CRISPR libraries were created from four different strains of *Toxoplasma*, with a representative strain from each lineage. Plus, the CRISPR screens were carried out in both mice and mouse-derived bone marrow macrophages. In both cases, GRA12 was identified as a virulence factor

across parasite strains. Interestingly, GRA12 had no effect on rat BMDMs or in human fibroblasts. While the experimental question is of interest to the field and a variety of techniques were used, there are a few points that need to be clarified.

1. There appears to be a discrepancy in the results of the plaque assays. For the VAND parasite strain, Δ GRA12 has a slight increase in plaque area compared to the parental or complemented strains (Fig. S2D). For the RH strain, Δ GRA12 has an increase in plaque area compared to the parental strain (as with VAND), but the complemented strain has an increase in plaque area compared to Δ GRA12 (Fig. S3D). For the VAND strain, the complemented strain mirrored the parental, which would be expected. But, for the RH strain, the complemented strain differed from the parental. Why the discrepancy? As GRA12 has a specific effect on mice versus rats (or human fibroblasts), it would be interesting to do plaque assays with MEFs instead of HFFs, especially as in the high-content imaging assay of PRU Δ GRA12 fewer parasites were detected in BMDMs and MEFs but not in HFFs (Fig S4D).

2. In the movie (and images) of IFN-gamma treated BMDM infected with RH Δ GRA25, there are two cells that appear to shrink and then burst. The cell on the left shrinks (no dark structures appear), parasites egress, and the cell bursts. The cell on the right shrinks, dark structures appear with some blebbing, and then the cell bursts. Is this cell infected? Why do dark structures and blebbing happen with this cell and not the one next to it? Is there more than one type of cell death happening?

3. For the EM images showing vacuolar collapse, the figure legend states, "TEM images of IFN-gamma treated BMDM or untreated PWD/Phj BMDM....", however, only the IFN-gamma treated images are shown. It would be helpful to show both. Plus, how was the vacuolar area measured in Fiji ImageJ? Was an ROI drawn around the vacuole? It would be useful to also include how many vacuoles had the "collapse" phenotype in the population for each condition. By IFA, are there any observable defects in the localization of the PVM protein GRA3 or in the vacuolar protein GRA2?

4. The localization of GRA12 to the intravacuolar space is interesting as is the hypothesis that GRA12 may play a role in stabilizing the vacuole. Was the IFA done with infected HFF? Does GRA12 have a slightly different localization in MEF or BMDM (either untreated or IFN-gamma treated)? GRA12 localization in these cells would be interesting as Δ GRA12 has more of a defect in them.

5. In Figure S6, the ability of host factors (IRG) to target the PV was measured for RH Δ GRA12 versus parental. The RH strain, however, does not show much recruitment of host factors (e.g., IRGb10) to the PV. While it is a good way to show that there is no increase in recruitment with Δ GRA12 parasites, it would be good to assess PRU Δ GRA12 parasites since the PRU strain has a more dramatic recruitment. For the representative images, which strain is shown? PRU or RH?

6. The GRA12 orthologues from *H. hammondi* and *N. caninum* localized within the vacuole and rescued the RH Δ GRA12 defect in restriction assays (Fig. 5D). However, in the plaque assays the *H. hammondi* orthologue was not able to complement the Δ GRA12 mutation. Is this a difference between BMDM and fibroblasts? Or, between mouse versus human cells? What happens if a plaque assay is done with MEFs?

7. Minor point - In the graphs Fig. 4A, 4F, S5A and S5B, it is difficult to distinguish.

Reviewer #3

(Remarks to the Author)

Toxoplasma gondii-derived secreted effectors are recognized as essential virulence factors in murine infections. These effectors extensively manipulate host signaling pathways and modulate host gene expression through direct or indirect interactions with host proteins. Such interactions are crucial for *T. gondii*'s ability to evade immune clearance and establish a favorable replicative niche within the host.

To date, most genes implicated in *T. gondii* virulence exhibit strain-specific differences in virulence phenotypes. In fact, most of these genes were initially identified using classical genetic approaches, such as genetic crosses between different *T. gondii* strains. Although this approach has successfully revealed many strain-specific effectors, it has been limited in identifying core virulence factors conserved across *T. gondii* strains. In this study by Torelli et al., the authors conducted a targeted CRISPR screen of 253 predicted rhoptry and dense granule proteins across four distinct strains of *T. gondii* and various murine subspecies. This screen identified GRA12 as a conserved core virulence factor among different *T. gondii* strains, with a function that is also conserved in closely related Coccidian parasites. The study suggests that GRA12 is vital for parasite survival within the host, potentially playing a role in mediating defense against IFN γ -driven host responses and preventing necrotic host cell death.

I believe this paper significantly advances our understanding of *Toxoplasma* biology. The findings are intriguing, well-supported, and contribute valuable knowledge to the field. However, I would like to see additional experiments aimed at uncovering the precise mechanism of GRA12 action. Considering the differences among *T. gondii* strains, such as variations in the expression of ROP5 and ROP18, I suggest conducting Co-IP experiments in both type II and type III strains to explore potential differences in GRA12 interactions.

I recommend performing co-immunoprecipitation with a modified protocol to identify potential interacting partners of GRA12.

Identifying these binding partners could provide insights into the mechanisms through which GRA12 protects the parasite's replicative niche. The role of GRA12 has been established through recent publications and it would strengthen the manuscript to have some additional functional data

Some minor issues

Line 138- 141: "Five days post infection, parasites were recovered from the peritoneal exudates, their sgRNAs amplified by PCR and sequenced to determine their relative abundance before and after in vivo selection"

Is there a specific reason why the parasites were cultured prior to DNA extraction? This approach differs from the previous lab's I protocol. has this approach improved the gRNA read counts?

Sangaré et al. demonstrated that there was no significant difference in gRNA abundance between in vitro-grown parasites and those directly extracted from the peritoneal cavity, suggesting that either method could be used without impacting results.

Version 1:

Reviewer comments:

Reviewer #1

(Remarks to the Author)

The authors should cite a study on GRA12 paralogues (Guevara et al., mSphere, 2021, PMID: 33883265) (Line 449). Apart from that, the authors have addressed all the concerns by providing additional data or comments.

(Remarks on code availability)

Reviewer #2

(Remarks to the Author)

The authors have fully answered each of my questions and comments. The work showing that GRA12 is a virulence factor across Toxoplasma lineages will be valuable and of interest to the field of parasitology.

(Remarks on code availability)

Reviewer #3

(Remarks to the Author)

well revised! thank you for addressing our comments

(Remarks on code availability)

Following the reviewers' suggestions, we performed new experiments that deserved to be integrated in the manuscript and required a rearrangement in the figure panels. In particular, Figures 5-6 and Supplementary Figures 6-9 and the relative Results sections were modified. We believe that this new order greatly improves the readability and value of the manuscript.

Reviewer #1 (Remarks to the Author):

In this manuscript by Torelli et al., the authors propose that GRA12 is a fundamental virulence gene independent of parasite strain or mouse subspecies. Most studies in *Toxoplasma* so far have been conducted in lab-adapted strains in parasites (e.g., RH and Pru) and mice (e.g., C57BL/6 and BALB/c). However, this does not necessarily represent real-world host-parasite interactions. The authors used a highly virulent atypical strain VAND, and mouse subspecies that possess polymorphisms in IRGs. The authors conducted multiple in vivo CRISPR screens and elegantly showed that ROP18 depends on parasite strains and mouse subspecies as expected. Interestingly, GRA12 is the top hit on any screens. Deletion of GRA12 results in host cell necrosis in unknown mechanisms but is partially dependent on the early egress of parasites. Although the authors do not fully address the function of GRA12, this paper should be of great interest to the field of *Toxoplasma* parasitology.

Major concerns

1. The authors previously designed a gRNA library based on the type II ME49 genome. They aligned gRNA sequences to the type I GT1 genome to validate that the gRNA library can be used for both strains (Young et al., Nat commun, 2019, PMID: 31481656). Did the authors validate the gRNA library for the type III and VAND genomes? This validation data must be shown in the supplemental data in the way of Young et al. (Nat commun, 2019, PMID: 31481656).

The reviewer raises an important point. We followed their suggestion by aligning the gRNA library, initially designed using the ME49 genome (ToxoDB, release 41), to the ME49, GT1, VAND and VEG genomes (ToxoDB, release 68). We describe the alignment analysis, which was performed in a similar fashion to the analysis reported in Young et al. in the methods section under "Library generation" and added the resulting information in Supplementary Data 2. In line with results from Young et al the overall conservation across sequences were between 96.92% (VAND) and 99.61% (ME49).

2. This reviewer needs clarification on Figure S5F. Are these samples treated or non-treated by IFN- γ , or just mislabeled?

We thank the reviewer for spotting this error due to mislabelling, as samples were untreated. We now corrected the legend of Figure S5F.

3. Line 408-412, the authors described that GRA12 paralogues (GRA12A-D) are not essential for parasite infection in vivo based on the results of published CRISPR

screens. However, there is a study that investigates GRA12 paralogues (Guevara et al., mSphere, 2021, PMID: 33883265). This paper showed that GRA12 paralogues are required for acute virulence of low-dose infection and cyst burden. Although this reviewer agrees that GRA12 is the most important for acute infection, this paragraph could be revised, and Guevara et al. should be cited.

The reviewer is correct that the current sentence undermines the *in vivo* relevance of the GRA12-paralogs, and we rephrased the sentence as follow: “We and others have shown that GRA12 has a key role for *Toxoplasma* survival *in vivo* ^{49–51}, while the paralogues GRA12A, GRA12B and GRA12D contribute to virulence only during mild infections and are linked to cyst formation.”

4. In Figure 2B, a single band of GRA12-HA is observed around 40 kDa. In Figure S3C, there is another band of GRA12-HA below 25 kDa in RH Δ GRA12::GRA12-HA. Is there also a smaller band in VAND Δ GRA12::GRA12-HA (Figure 2A)? What is the smaller band of GRA12-HA?

In all performed western blots of HA-tag GRA12 lysates, regardless of the strain VAND or RH, or whether it was endogenously tagged or in the complemented version, a band around 25 kDa was observed. For the specific blot in figure 2B, the membrane was cut at 30 kDa to blot it with different antibodies at the same time, however we have complete membranes with the same VAND Δ GRA12::GRA12-HA lysate showing the band at 25 kDa. A band of similar size was previously seen in Michelin et al. (PMID: 18840447) using anti-GRA12 antibodies, similarly to our blots with anti-HA antibodies. This suggests that this band is a product of GRA12, either as a result of degradation or processing, which we did not further investigate here.

Minor comments

1. In Figure S1A, how did the authors calculate the transfection efficiency of each strain? I assume these are GFP-positive rates of parasites, but the details should be described in the material and methods.

The transfection efficiency was calculated from the survival rate of stably transfected parasites established by plaque assay, and the details have now been added to the section “CRISPR pool creation” of Materials and Methods.

Reviewer #2 (Remarks to the Author):

Toxoplasma gondii strains are grouped into lineages (I, II, III, and atypical) that have different levels of virulence in mice and between mouse strains. Taking a novel approach, the authors have designed CRISPR screens to identify virulence factors that are conserved across parasite strains. A strength of the paper is that the CRISPR libraries were created from four different strains of *Toxoplasma*, with a representative strain from each lineage. Plus, the CRISPR screens were carried out in both mice and mouse-derived bone marrow macrophages. In both cases, GRA12 was identified as a virulence factor across parasite strains. Interestingly, GRA12 had no effect on rat BMDMs or in human fibroblasts. While the experimental question is of interest to the

field and a variety of techniques were used, there are a few points that need to be clarified.

1. There appears to be a discrepancy in the results of the plaque assays. For the VAND parasite strain, Δ GRA12 has a slight increase in plaque area compared to the parental or complemented strains (Fig. S2D). For the RH strain, Δ GRA12 has an increase in plaque area compared to the parental strain (as with VAND), but the complemented strain has an increase in plaque area compared to Δ GRA12 (Fig. S3D). For the VAND strain, the complemented strain mirrored the parental, which would be expected. But, for the RH strain, the complemented strain differed from the parental. Why the discrepancy?

The reviewer is correct in highlighting discrepancies in complemented strain plaque sizes between VAND and RH strains, both in point 1 and 6 of their revision. Because all complemented strains complement the tested phenotypes under IFN γ conditions, we reason that these differences in growth are unrelated to GRA12, but probably caused by a secondary mutation in the strain. We note that plaque assays are very sensitive to small differences in growth between strains over the course of time used here (i.e. 10-14 days), and that differences in growth are not readily observed in restriction assays or normal culturing conditions (i.e. 1-2 days). However, we felt it was important to show that these small differences in growth under non-IFN γ conditions occur, in case other people use this line in the future. To note, we decided to remove the statistics from all plaque assay analysis in the manuscript, since the number of biological repeats was below 3 and we prefer to consider individual plaques as technical replicates rather than biological replicates.

As GRA12 has a specific effect on mice versus rats (or human fibroblasts), it would be interesting to do plaque assays with MEFs instead of HFFs, especially as in the high-content imaging assay of PRU Δ GRA12 fewer parasites were detected in BMDMs and MEFs but not in HFFs (Fig S4D).

This is a good suggestion, and we repeated the plaque assays in MEFs with all established RH strains and the VAND strains. Data have now been integrated in the Supplementary Figures throughout the manuscript. Overall, we observed the same growth pattern between the two cell types, supporting the hypothesis that secondary mutations influence the strain fitness in long term growth studies, but not in short term IFN γ assays. We also observed that all strains had remarkably smaller plaques in MEFs compared to HFFs. The nature of this difference is not known and is beyond the scope of this study, but is now specifically mentioned.

2. In the movie (and images) of IFN-gamma treated BMDM infected with RH Δ GRA25, there are two cells that appear to shrink and then burst. The cell on the left shrinks (no dark structures appear), parasites egress, and the cell bursts. The cell on the right shrinks, dark structures appear with some blebbing, and then the cell bursts. Is this cell infected? Why do dark structures and blebbing happen with this cell and not the one next to it? Is there more than one type of cell death happening?

We analysed the morphological changes in the live cell imaging and observed that two main phenotypes were evident: the shrinkage followed by an explosive burst like in the two central cells in the video, and the less pronounced shrinkage with blebbing and nucleus condensation as in the cell in the bottom left corner (both phenotypes now highlighted in the figure). We observed that both phenotypes were present in either parental, KO and complemented strains, and observed that Δ GRA12 parasites just show a general increase in the absolute number of both bursting and blebbing cells compared to the other strains following IFN γ treatment. Within both conditions we observed variability (e.g. presence/absence of dark granules) and egressing parasites were observed often but not always, probably because of the interval of in the time-lapse.

Since both phenotypes were observed equally in all strains, we now changed the sentence at line 331 to better represent the findings “Live time-lapse microscopy of IFN γ -treated BMDMs infected with parasites showed two main cell death phenotypes: a sudden shrinkage followed by a “burst”, or blebbing with nuclear condensation, with parasites sometimes egressing from dying cells. Both phenotypes appeared regardless of the strain used (Figure 4d, Supplementary Movie 1).”

3. For the EM images showing vacuolar collapse, the figure legend states, “TEM images of IFN-gamma treated BMDM or untreated PWD/Phj BMDM....”, however, only the IFN-gamma treated images are shown. It would be helpful to show both. Plus, how was the vacuolar area measured in Fiji ImageJ? Was an ROI drawn around the vacuole? It would be useful to also include how many vacuoles had the “collapse” phenotype in the population for each condition. By IFA, are there any observable defects in the localization of the PVM protein GRA3 or in the vacuolar protein GRA2? We now added examples of TEM images of untreated infected BMDMs in Supplementary Figure 7c, as for space reasons it did not fit in the main figure, as well as the numbers of collapsed vacuoles in Figure 5d. The vacuolar area was calculated in Fiji as explained in Materials and Methods “Transmission Electron Microscopy”, and an example of the drawn ROIs is shown in Figure 5c.

We confirmed the localisation by IFA of GRA3 and GRA2 in Δ GRA12 vacuoles as opposed to the parental strain, and did not observe abnormalities. Additionally, we exclude that GRA3 and GRA2 are related to GRA12 phenotype, since they don't display a negative score in any screen of our group or others. This additional experiment has been included in Supplementary Figure 6.

4. The localization of GRA12 to the intravacuolar space is interesting as is the hypothesis that GRA12 may play a role in stabilizing the vacuole. Was the IFA done with infected HFF? Does GRA12 have a slightly different localization in MEF or BMDM (either untreated or IFN-gamma treated)? GRA12 localization in these cells would be interesting as Δ GRA12 has more of a defect in them.

As the IFAs in the manuscript were performed in HFFs, we followed the reviewer's advice and repeated the experiment in BMDMs. No visible differences in localisation

were observed with or without IFN γ . This additional experiment has been included in Supplementary Figure 6.

5. In Figure S6, the ability of host factors (IRG) to target the PV was measured for RH Δ GRA12 versus parental. The RH strain, however, does not show much recruitment of host factors (e.g., IRGb10) to the PV. While it is a good way to show that there is no increase in recruitment with Δ GRA12 parasites, it would be good to assess PRU Δ GRA12 parasites since the PRU strain has a more dramatic recruitment. For the representative images, which strain is shown? PRU or RH?

This was a good suggestion, and we performed IFAs in type II PRU, which induces higher recruitment of host factors compared to the RH strain, and integrated the results in Figure 5 and Supplementary Figure 8. We observed that PRU Δ GRA12 parasites significantly recruit 2-6 fold less of IRGd, IRGb10 and GBP2 to their PVM compared to the control PRU Δ UPRT strain. These results suggest that GRA12 affects the PVM composition and recognition from the host immune system. We modified the Result and Discussion sections, as well as the representative images, to include these findings. Despite the low levels of recruitment for type I parasites, the same results were observed in RH for IRGd and IRGb10 (now with statistical analysis in Supplementary Figure 8).

6. The GRA12 orthologues from *H. hammondi* and *N. caninum* localized within the vacuole and rescued the RH Δ GRA12 defect in restriction assays (Fig. 5D). However, in the plaque assays the *H. hammondi* orthologue was not able to complement the Δ GRA12 mutation. Is this a difference between BMDM and fibroblasts? Or, between mouse versus human cells? What happens if a plaque assay is done with MEFs?

This point, similarly to point 1, raises a fitness discrepancy of the different complemented strains. We invite the reviewer to go to point 1 of the rebuttal where we addressed this concern.

7. Minor point - In the graphs Fig. 4A, 4F, S5A and S5B, it is difficult to distinguish. We now increased the size of these graphs.

Reviewer #3 (Remarks to the Author):

Toxoplasma gondii-derived secreted effectors are recognized as essential virulence factors in murine infections. These effectors extensively manipulate host signaling pathways and modulate host gene expression through direct or indirect interactions with host proteins. Such interactions are crucial for *T. gondii*'s ability to evade immune clearance and establish a favorable replicative niche within the host.

To date, most genes implicated in *T. gondii* virulence exhibit strain-specific differences in virulence phenotypes. In fact, most of these genes were initially identified using classical genetic approaches, such as genetic crosses between different *T. gondii*

strains. Although this approach has successfully revealed many strain-specific effectors, it has been limited in identifying core virulence factors conserved across *T. gondii* strains. In this study by Torelli et al., the authors conducted a targeted CRISPR screen of 253 predicted rhoptry and dense granule proteins across four distinct strains of *T. gondii* and various murine subspecies. This screen identified GRA12 as a conserved core virulence factor among different *T. gondii* strains, with a function that is also conserved in closely related Coccidian parasites. The study suggests that GRA12 is vital for parasite survival within the host, potentially playing a role in mediating defense against IFN γ -driven host responses and preventing necrotic host cell death.

I believe this paper significantly advances our understanding of *Toxoplasma* biology. The findings are intriguing, well-supported, and contribute valuable knowledge to the field. However, I would like to see additional experiments aimed at uncovering the precise mechanism of GRA12 action. Considering the differences among *T. gondii* strains, such as variations in the expression of ROP5 and ROP18, I suggest conducting Co-IP experiments in both type II and type III strains to explore potential differences in GRA12 interactions.

I recommend performing co-immunoprecipitation with a modified protocol to identify potential interacting partners of GRA12. Identifying these binding partners could provide insights into the mechanisms through which GRA12 protects the parasite's replicative niche. The role of GRA12 has been established through recent publications and it would strengthen the manuscript to have some additional functional data

Since we showed that GRA12 function is conserved across strains, we do not see the potential benefit of creating novel GRA12-HA strains in type II and type III to identify interaction partners. In our pull-down protocol we reproduced conditions where GRA12 displayed a phenotype (IFN γ -treated BMDMs) and used a NP-40 buffer to solubilise the membrane-bound part of the protein, as highlighted by Michelin et al (PMID: 18840447).

Instead of trying different IP conditions, as an alternative approach, we established a C-terminally TurboID-tagged strain to identify the proximal environment of GRA12 via biotin treatment and mass spectrometry. The data (explained in detail below), show that GRA12 has specific proteins in its vicinity in intracellular parasites (after secretion), compared to extracellular parasites (when GRA12 is in the dense granules prior to secretion). However, none of the identified proteins shows a fitness defect in the CRISPR screens. This excludes these as probable important partners for GRA12 function, unless of course there is redundancy. Since these negative data do not contribute significantly to the manuscript and are part of future work of the lab, we

would like not to integrate them as publicly available data at the moment. However, for the reviewer to evaluate the results- we describe the experiment in detail below.

Details of the experiment. We complemented the RH Δ GRA12 strain, by expressing GRA12 fused with the biotin ligase TurboID and a V5 tag in the *Uprt* locus. GRA12 localisation mirrored what was observed in previously established tagged strains (Fig 1A), showing biotin ligase activity in the same locus (Fig 1B), and rescue of the IFN γ restriction of parental Δ GRA12 parasites indicating that the fusion protein is functional (data now shown). By western blot, the fused GRA12-TurboID-V5 protein migrated as expected at 74 KDa (Fig 1C) and numerous proteins were identified via anti-streptavidin staining, supporting the protein activity. Macrophages pre-treated with IFN γ were infected with either the control RH Δ Ku80 (WT) strain or the RH Δ GRA12::GRA12-TurboID-V5 (GRA12-TurboID) strain for 24h before pulsing with 50 μ M biotin for 90 minutes. Additional controls of biotin pulsed, syringe-lysed and purified GRA12-TurboID parasites were included to discriminate between the intracellular and extracellular proximity environments of GRA1 (Fig 1D). The comparison of intracellular vs purified extracellular parasites can effectively identify secreted dense granule proteins (see Treeck et al., 2011, Cell Host and Microbe). Following lysis and pull-down via streptavidin binding, samples were used for mass spectrometry and analysed. Several proteins were highly biotinylated in the intracellular TurboID samples in contrast to background in WT controls, suggesting a close proximity with GRA12 (Fig 1E). Proteins with the highest enrichment includes the dense granule proteins GRA2, MAG1 and GRA7, as well as not secreted proteins as SAG1 and the ER-associated TGGT1_229480 (Fig 1E). When considering the enrichment in the intracellular samples compared to extracellular samples, we could distinguish two populations: one with a lower enrichment score comparable to GRA12 composed of eg. GRA1, GRA7 and GRA2, and one with a higher score with proteins like GRA15, GRA42 and TGGT1_200360 (Fig 1F). A few host proteins like alpha tubulin and the mitochondrial USMG5 were identified in the latter population, supporting a PVM localisation of this group. Importantly, none of the top hit proteins identified in this experiment displayed a negative fitness score in our or other screens, which we would expect from a protein acting together with GRA12.

Figure 1. Proximity labelling of GRA12 does not identify putative protein partners. Immunofluorescence localisation of A) the C-terminal V5-tagged GRA12-TurboID protein in the RH Δ GRA12::GRA12-TurboID (GRA12-TurboID) strain compared to parental RH Δ GRA12 strain, and B) the localisation of the biotinylated proteins via streptavidin AF488-conjugated staining. Parasites express an mCherry cytosolic reporter and the scale bar is 10 μ m. C) Western blot analysis of streptavidin pull-downs from HFF 150 μ M biotin 90 min. Lanes: RH Δ KU80 and RH Δ GRA12::GRA12-TurboID-V5. Molecular weight markers (KDa) are shown on the left. Bands for GRA12-TurboID-V5, streptavidin, and IRDye 800CW are indicated. Bottom: Anti-*T. gondii* IRDye 680CW blot showing parasite loading control.

Western blot of lysates of parental RH Δ Ku80 (WT) and GRA12-TurboID parasites pulsed with 150 μ M biotin for 90min in HFF cultures. The membrane was blotted with streptavidin conjugated with IRDye 800CW, and an anti-Toxoplasma antibody was used as loading control. D) Scheme of the mass spectrometry experiment. PWD/PhJ BMDMs were pretreated for 24h with 10U/ml IFN γ before infection with WT or GRA12-TurboID parasites for 24h in triplicate. Samples were pulsed with 50 μ M biotin for 90min before lysis and pull-down with streptavidin-conjugated agarose beads and analysed via mass spectrometry. An additional sample of GRA12-TurboID parasites was syringe-lysed and purified through PD10 columns before pull-down, to represent the extracellular GRA12 neighbouring proteome control. E) Volcano plot of the Log₂ fold change (L2FC) of biotin enrichment of proteins in GRA12 proximity in GRA12-TurboID vs WT samples. F) Scatter Plot of proteins in GRA12 proximity in GRA12-TurboID vs WT samples, in relation to how enriched they are in the intracellular vs extracellular populations.

Some minor issues

Line 138- 141: “Five days post infection, parasites were recovered from the peritoneal exudates, their sgRNAs amplified by PCR and sequenced to determine their relative abundance before and after in vivo selection”

Is there a specific reason why the parasites were cultured prior to DNA extraction? This approach differs from the previous lab's I protocol. has this approach improved the gRNA read counts?

Sangaré et al. demonstrated that there was no significant difference in gRNA abundance between in vitro-grown parasites and those directly extracted from the peritoneal cavity, suggesting that either method could be used without impacting results.

There are several reasons why we decided to perform the amplification step in HFFs prior to sequencing: 1) Sangare et al. used 10 guides/target and observed that regardless of the method (straight sequencing vs amplification) they remained with an average of 5 guides/target. We instead used 5 guides/target and set a strict threshold of minimum 3 guides/target in our analysis, which was not compatible with a possible 50% loss. 2) We wanted to compare these screens with those previously performed in our lab (Young et al, Butterworth et al and Lockyer et al) where also an amplification step was involved. 3) We expected a higher in vivo restriction for certain *Toxoplasma* strains (VEG) and mouse subspecies (CAST/EiJ and PWD/PhJ).

We thank all reviewers for their feedback and suggestions which greatly improved the manuscript and for their kind words following revision.

Reviewer #1 (Remarks to the Author):

The authors should cite a study on GRA12 paralogues (Guevara et al., mSphere, 2021, PMID: 33883265) (Line 449).

Apart from that, the authors have addressed all the concerns by providing additional data or comments.

We thank reviewer 1 for highlighting the absence of this important and appropriate citation, which was likely erroneously removed in subsequent version of the manuscript. We now included the citation in the final version.

Reviewer #2 (Remarks to the Author):

The authors have fully answered each of my questions and comments. The work showing that GRA12 is a virulence factor across Toxoplasma lineages will be valuable and of interest to the field of parasitology.

Reviewer #3 (Remarks to the Author):

well revised! thank you for addressing our comments